# The RNA-binding KH-domain in the unique transcription factor of the malaria parasite is responsible for its transcriptional regulatory activity

Kanako Komaki-Yasuda◯*, Shigeyuki Kano

Research Institute, National Center for Global Health and Medicine, Shinjuku-ku, Tokyo, Japan

* komaki@ri.ncgm.go.jp

## Abstract

To date, only a small number of regulatory transcription factors have been predicted from the genome of *Plasmodium* and Apicomplexan parasites. We previously identified a nuclear factor named Prx regulatory element-binding protein (PREBP) from *Plasmodium falciparum*. PREBP had been suggested to bind to the *cis*-element in the promoter of an antioxidant *pf1-cys-prx* gene, thereby promoting the expression of downstream genes. PREBP has 4 putative K homology (KH) domains, which are known to bind RNA and single-stranded DNA. In this study, to understand the detailed action of PREBP in parasite cells, we first observed that in living parasite cells, PREBP was localized in the nucleus in the trophozoite and schizont stages, in which the expression of the target *pf1-cys-prx* was enhanced. The interaction of PREBP and the *cis*-element of *pf1-cys-prx* in the parasite cells was also confirmed. Further, the activities of PREBP deletion mutants were analyzed, and regions with repeated KH domains in PREBP seemed to be responsible for the recognition of the *cis*-element. These results led us to hypothesize that *Plasmodium* and other Apicomplexan parasites might have a transcription factor family with KH domains. Bioinformatic analysis revealed a putative ortholog group including PREBP and several *Plasmodium* and Apicomplexan factors with KH domains. One of the *P. falciparum*-derived factors, which were included in the putative ortholog group, was found to be localized at the nucleus in the trophozoite stage, indicating that it might be a novel transcription factor. The discovery of PREBP and putative transcription factors with KH domains suggested that multi-functional proteins with KH domains possibly evolved in the Apicomplexan organisms. They might play key roles in transcriptional regulatory processes that are essential for living organisms and may even represent unique drug targets for malaria therapy.

## Introduction

Malaria, the world's most important life-threatening parasitic infectious disease, is caused by protozoan parasites of the genus *Plasmodium* [1]. At the time of writing this report, there is no effective vaccine, and the rapid emergence of drug-resistant parasites is making malaria

**Data Availability Statement:** All relevant data are included within the manuscript and its Supporting information Files. Biological materials are available upon request, subject to the appropriate material

transfer agreement. Requests for the materials should be submitted to: Intellectual Property and Contract Management Office, National Center for Global Health and Medicine (Email: chizai_keiyaku@hosp.ncgm.go.jp).

**Funding:** This work was supported by JSPS (Japan Society for the Promotion of Science) KAKENHI Grant Numbers 17K08817 and 20K07473 (to KKY). The funder's URL: https://www.jsps.go.jp/. This work was also supported by AMED (Japan Agency for Medical Research and Development) under Grant Number 23jk0210006h0001 (to SK). The funder's URL: https://www.amed.go.jp. The funders had no role in study design, data collection and analysis, decision to publish, or preparation of the manuscript.

**Competing interests:** The authors have declared that no competing interests exist.

difficult to control [2]. For the development of new strategies to combat malaria, a better understanding of the basic biology of this protozoan parasite is required. The mechanisms underlying the regulation of its gene expression remain mostly unclear, and only a small number of regulatory transcription factors have been predicted in the *Plasmodium* genome [3, 4]. A bioinformatics analysis of the genomes of *Plasmodium* and several Apicomplexan parasites revealed that the hypothetical ApiAP-2 family with plant AP-2-like DNA-binding domains was the gene family that encoded transcription factors [5, 6]. In the genome of *P. falciparum*, 26 genes were predicted to encode ApiAP-2 putative transcription factors. Some members of the ApiAP-2 family have been revealed to be functional transcription factors, which regulate gene expression in the gametocyte [7–10], ookinete [11, 12], sporozoite [13], and hypnozoite [14] stages. In the intraerythrocytic stage, an ookinete-specific transcription factor, AP2-O, has recently been suggested to be implicated in the expression of the virulent PfEMP1 gene family [12]. However, the detailed functions of transcription factors in the intraerythrocytic cell cycle remain to be elucidated.

In terms of the general transcriptional regulatory mechanisms in *P. falciparum*, we are interested in the mechanism that regulates the transcription of the *pf1-cys-prx* gene (PF3D7_0802200) as a model for understanding the stage-specific gene expression. *pf1-cys-prx* is a gene for an antioxidant protein belonging to the peroxiredoxin family [15], and the expression of *pf1-cys-prx* is specifically elevated during the trophozoite/schizont stages [16]. In a previous study, we precisely analyzed the promoter activity of the 5′ region of *pf1-cys-prx* and verified the 102-bp *cis*-acting enhancer region in this gene [17]. In addition, the *cis*-enhancer element was revealed to be a target of histone acetylation at the same timing as the elevation of transcription. Then, we purified and identified a nuclear factor, which bound specifically to the 102-bp *cis*-enhancer of the *pf1-cys-prx* gene, from the parasite nuclear extracts and named this factor Prx regulatory element-binding protein (PREBP) [18]. We showed that PREBP could specifically interact with the DNA sequence of the *cis*-enhancer of the *pf1-cys-prx* gene by an *in vitro* electrophoresis mobility shift assay. We also showed that PREBP could affect the promoter sequence of the *pf1-cys-prx* gene and promote the expression of the downstream reporter gene on plasmids transiently transfected into parasite cells. Thus, PREBP may be the key molecule for the regulation of the stage-specific expression of *pf1-cys-prx*.

The secondary structure of PREBP was predicted from its amino acid sequence, and the characteristic feature was that PREBP contained 4 putative K homology (KH) domains. The KH domain is known as a domain of approximately 70 amino acids in size, including various RNA or single-stranded DNA binding proteins that perform a wide range of cellular functions [19–21]. In humans, a transcription factor named FUBP1, which possesses 4 KH domains, was reported. FUBP1 regulates the expression of the *c-myc* gene by binding to the single-stranded far upstream element (FUSE), a *cis*-A/T-rich element located at 1.7 kb upstream of the *c-myc* open reading frame (ORF) [22–25]. The putative orthologs of PREBP are well-conserved in other *Plasmodium* species and Apicomplexan parasites [18]. These findings suggest the existence of transcription factors without the ApiAP-2 domain in Apicomplexan organisms. These situations had led us to question whether the deduced KH domains were indeed essential domains for the transcription factor activity of PREBP.

In this study, to understand the detailed action of PREBP in living parasite cells, we observed the cellular localization of PREBP throughout the intraerythrocytic developmental stages and then analyzed the interaction of PREBP with the promoter region of the *pf1-cys-prx* gene in parasite cells. Furthermore, we analyzed the functions of the KH domains of PREBP in relation to the recognition of the *cis*-enhancer element of *pf1-cys-prx* in the living parasite cell using PREBP deletion mutants. We also searched the genome databases of various

Apicomplexan parasites to find other proteins, including hypothetical KH domains for putative candidate transcription factors.

## Materials and methods

### Parasite

The FCR-3 strain of *P. falciparum* was cultured with the modified method of Trager and Jensen [26] in RPMI medium (Thermo Fisher Scientific, Waltham, MA, USA) supplemented with 10% human plasma. Tight synchronization of parasites was achieved by repetitive 5%-D-sorbitol treatment [27].

### Plasmid construction

The sequences of the primers used for plasmid construction are shown in S1 Table. For the construction of p1-10R PREBP ΔC2, ΔC3, ΔN1, ΔN2, and ΔN3, we used PCR reactions that amplify almost the whole region of the template plasmid, p1-10R PREBP [18], which only lacks the deleted part of the ORF of the PREBP gene. For p1-10R PREBP ΔC2 andΔC3, the following primer pairs were used, respectively: NotI-HSP86 3'F and ΔC2F, or ΔC3F. For p1-10R PREBPΔN1, ΔN2, and ΔN3, the following primer pairs were used, respectively: NotI-FLAGR and ΔN1F, ΔN2F, or ΔN3F. These primers contain a *Not*I restriction site at their 5′ terminal region. The amplified fragments contain most of the template plasmid and lack only part of the 5′ or 3′ terminal region of the coding sequence of PREBP. After PCR, the amplified fragments were treated with DpnI to digest the template plasmid and were treated with *Not*I to obtain a *Not*I cut site on both sides of the fragment. Then, the fragments were treated with T4 ligase for self-ligation to make the circular form, which results in the plasmid. The ligation reaction mixture was then transfected into *E. coli*, and plasmid DNA was collected and confirmed by a DNA sequencing analysis. For the construction of p1-10R PREBP ΔC2 (-*cis*-enhancer) and p1-10R PREBP ΔC3 (-*cis*-enhancer), the same strategy was used as for the construction of p1-10R PREBP ΔC2 and p1-10R PREBP ΔC3, except for the use of the basal p1-10R PREBP (-*cis*-enhancer) as the template for the initial PCR assay. For the construction of p1-10R PREBPΔC1, however, we used a different strategy from the other plasmids, i.e., the same strategy used for the construction of p1-10R PREBP in a previous study [16]. In brief, the partial sequence of the PREBP gene was amplified using the following primer pair with the p1-10R PREBP plasmid as a template: Xho-FLAG-PREBP-F and ΔC1-R. The amplified fragment contained *Xho*I sites at both the 5′ and 3′ ends. At first, to obtain the overexpression cassette for PREBP ΔC1, the fragments were inserted into the *Xho*I site of the pHC1 plasmid [28]. This plasmid was termed pHC1-PREBP-ΔC1. Then, PREBP-ΔC1 expression cassettes were excised from pHC1-PREBP-ΔC1 by *Hind*III and inserted into the same restriction site of p1-10R to make p1-10R PREBP ΔC1. In every step, the direction of the inserted fragments was checked by sequencing analyses. To establish the transgenic parasite lines that express PREBP-GFP and PF3D7_0302800-GFP fusion proteins under the control of the endogenous 5′ promoter, the plasmids pHC1-ΔPREBP-GFP and pHC1-ΔPF3D7_0302800-GFP were constructed (S1A Fig). To construct these plasmid vectors, we firstly amplified the C-terminal partial sequence of PREBP and PF3D7_0302800 by PCR using parasite genomic DNA extracted from cultured FCR-3 strain as a template. The following primer pairs were used for the amplification of the partial sequence of PREBP: Xho-PREBP-F and Not-PREBP-R. For the amplification of the PF3D7_0302800 partial sequence, the following primers were used: Xho-3D7_0302800-F and Not-3D7_0302800-R. The GFP gene was amplified by 2-step PCR. In the 1st PCR reaction, amplification was performed using the following primers with the pHRP GFPM2 plasmid [29] as template DNA: PSPT-GFP-F and Xho-GFP-R. The primer PSPT-GFP-F contains the

coding sequence for the peptide (Leu-Glu-Val-Leu-Phe-Gln-Gly-Pro) targeted for PreScission Protease and the N terminal sequence of the GFP gene. Then, the 2nd PCR assay was performed with the following primers and the 1st PCR product as the template: Not-PSPT and Xho-GFP-R and. Thus, PREBP, PF3D7_0302800, and the GFP sequence obtained by each PCR reaction contain *Not*I and *Xho*I sites at both ends of the amplicon sequence. The PCR-amplified partial sequence of PREBP or PF3D7_0302800 and the GFP sequence were digested with *Not*I and *Xho*I and ligated with the *Xho*I digested pHC1 vector. Finally, the fused sequence of the partial PREBP or PF3D7 and GFP was inserted into the *Xho*I site of the pHC1 vector.

## Establishment of transgenic parasite lines

Parasites were transfected with the pHC1-ΔPREBP-GFP or pHC1-ΔPF3D7_0302800-GFP plasmid by the same electroporation method used in the luciferase assay. At 48 h after transfection, 0.1 μM of pyrimethamine was added to the medium. During the 10 days after transfection, the medium was changed daily to remove dead parasites. Then, the medium was changed every two days until the growth of the transgenic parasite could be confirmed. After one month of culture, the integration of the vector sequence into the chromosome was confirmed by PCR. Then, the parasite was cloned by limiting dilution. The vector integration in each clone was again confirmed by PCR (S1B and S1C Fig). Fluorescence of GFP from the recombinant parasites was confirmed by fluorescence microscopy (S1D Fig).

## Observation of transcription factor and GFP fusion proteins in living parasite cells by confocal microscopy

Before the observation, the parasite culture was transferred to the 35-mm glass-bottom dish, then, 2 μg/mL of Hoechst 33342 was added directly to stain the parasite nucleus. Living parasite cells were observed as single-slice images using a confocal laser scanning microscope (FV3000, EVIDENT, Tokyo, Japan) under the 407-nm emission for the detection of Hoechst and under the 488-nm emission for the detection of GFP.

## Quantification of colocalization

The colocalization between GFP and the nucleus in each parasite cell was quantified using ImageJ (version 2.9.0/1.53t, National Institutes of Health, Bethesda, USA) [30]. Pearson's R value, indicating colocalization between the nucleus (stained with Hoechst, blue) and GFP (green), was calculated for each parasite cell image using the Coloc2 plugin in ImageJ [31]. An R value of 0 indicates the absence of colocalization, an R value of 1 represents 100% colocalization between the two colors, while an R value of -1 suggests incomplete colocalization.

## Chromatin immunoprecipitation assay

The chromatin immunoprecipitation (ChIP) assay was performed as specified in the ChIP Assay Kit (Cat No. 17295, Merck-Millipore, Burlington, USA) with minor modifications. In brief, $1–10 \times 10^9$ trophozoites (within 28–32 h after RBC infection) were isolated from infected erythrocytes by treatment with PBS containing 0.05% saponin. Parasites were washed several times with PBS and incubated overnight in culture medium containing 1% formaldehyde at 4°C. Cells were collected after two washes with PBS. Parasites were resuspended in 1000 μl of SDS lysis buffer. The mixture was sheared in a Handytype sonicator (TOMY, Tokyo, Japan [output control, level 7; 20 s on, 60 s off]) for 15 pulses to obtain DNA fragments in the range of 200–1000 bp. To remove debris, the mixture was centrifuged at $20,000 \times g$ for 10 min at 4°C,

and the supernatant termed the "chromatin supernatant", was served. Ten microliters of the chromatin supernatant (equivalent to $1–10 \times 10^7$ trophozoite lysates) was used as the input material for the following experiment. Before immunoprecipitation, 100 μl (equivalent to 1–-$10 \times 10^8$ trophozoite lysates) of chromatin supernatant was diluted in 1 ml of ChIP dilution buffer containing 1 mM PMSF (total of 10 tubes). Before immunoprecipitation, the chromatin sample was treated with 0.75 mg of protein A Dynabeads (Thermo Fisher Scientific) to absorb non-specific binding. Twenty-five micrograms of the rabbit polyclonal antibody against recombinant-PREBP [18] was added to the chromatin sample and incubated on a rotator for 1 h at 4˚C. The antigen-antibody complex was allowed to bind to 0.75 mg of protein A Dynabeads for 1 h at 4˚C. The beads were collected on a magnet stand, washed once with Low Salt Immune Complex Wash Buffer, once with High Salt Immune Complex Wash Buffer, and once with LiCl Immune Complex Wash Buffer, and twice with TE (10 mM Tris-HCl, pH 8.0 and 1 mM EDTA). The antigen-antibody complex was released from the beads by incubation in 400 μl of 1% SDS with 100 mM $NaHCO_3$ for 10 min at room temperature. The eluate and input starting material were heated in the presence of 200 mM NaCl at 65˚C for 4 h to reverse the formaldehyde cross-link. The samples were then digested with 0.1 mg/ml proteinase K at 45˚C for 1 h and subjected to phenol-chloroform extraction. The DNA was recovered using a FASTGene PCR purification Kit (Nippon Genetics, Tokyo, Japan) and eluted in 20 μl of TE. This DNA was analyzed by quantitative real-time PCR with exactly the same method described in a previously published paper [17]. Confirmation of the presence of PREBP in the eluate from Dynabeads through Western blotting validated the suitability of the anti-PREBP antibody for ChIP (S2 Fig).

## Luciferase assay

For the luciferase assay, plasmids were transfected into *P. falciparum* as described previously [17]. In brief, parasites ($3–10 \times 10^7$) synchronized at the ring stage were transfected with 50 μg of each plasmid DNA at 0.310 kV and 975 μF in a 0.2-cm gap cuvette with Gene Pulser II (Bio-Rad, Hercules, CA). After electroporation, parasites were transferred into a culture dish with 20 ml of medium and red blood cells at a final hematocrit of 2%. Each transfection cuvette contained 50 μg of pHC1-Rluc, a plasmid expressing Renilla luciferase, as a control for efficiency and recovery (dual-luciferase assay). Then, the parasite culture was harvested 20–24 h after transfection. Each culture was lysed for 5 min at room temperature with 2 ml of PBS containing 0.15% saponin. Parasite cells were washed twice in PBS and suspended in 50 μl of passive lysis buffer (Promega, Fitchburg, WI, USA). Firefly and Renilla luciferase activities in parasite extracts were analyzed with a Dual-Luciferase Reporter Assay System (Promega, Madison, WI) and Turner luminometer according to the manufacturer's instructions. The firefly luciferase activity was normalized to the Renilla luciferase activity, which was raised from the standard pHC1-Rluc transfection.

## Statistical analysis

Differences were evaluated with the Student *t*-test. *P* values of $< 0.01$ were considered to be statistically significant.

## Bioinformatics analysis

To search for candidate proteins that contain KH-domains in Apicomplexan organisms, we first screened the genome database for each *Plasmodium* (PlasmoDB 28, release 30 March 2016) [32], *Toxoplasma* (ToxoDB 28, release 30 March 2016) [33], and *Cryptosporidium* (CryptoDB 28, release 30 March 2016) [34] by the word "KH". Subsequently, proteins from the *P. falciparum* 3D7 strain, *T. gondii* TG1 strain, and *C. parvum* Iowa II strain, identified through this search, were selected for further analysis.

To define the protein domain organization of the detected candidate proteins, sequences were subjected to domain profiling using the Simple Modular Architecture Research Tool (SMART) [35]. Then, the sequences of candidate proteins and human FUBP1 (GenBank: AAH17247.1) were subjected to the BLASTClust program [36] to cluster them into putative ortholog groups. The "Sequence length to be covered" and "Percentage identity threshold" parameters were set to 20% and 20%, respectively. To gain greater insight into the similarity relationship between each candidate protein, every candidate sequence was subjected to a BLASTP search against each genome database for *P. falciparum*, *T. gondii*, and *C. parvum*. Proteins from the *P. falciparum* 3D7 strain, *T. gondii* TG1 strain, and *C. parvum* Iowa II strain, identified through this search, were selected. For the BLASTP search, the exception value, the maximum descriptions/alignments (V = B), and the low complexity filter were set at 10, 50, and "no", respectively (default settings).

## Results

### Cellular localization of PREBP through intraerythrocytic development of the parasite

To observe the cellular localization of PREBP, a transgenic *P. falciparum* parasite line was established. In the transgenic parasites, a GFP coding DNA sequence was integrated into the PREBP locus on the chromosome; thus, the fusion protein of PREBP and GFP was expressed under the control of the endogenous PREBP promoter (S1 Fig). Then, we observed the cellular localization of PREBP-GFP by confocal laser microscopy (Fig 1). In the ring stage, PREBP-GFP was localized outside of the nucleus. During the trophozoite stage, when green fluorescence derived from PREBP-GFP was observed throughout the cytoplasm, a particularly strong region of green fluorescent localization was observed, which coincided with the location of nuclear localization. Then, in the early schizont stage, the localization position of the green fluorescence of PREBP-GFP coincided with the nuclear position. During the late schizont stage, green fluorescence corresponding to the localization of PREBP-GFP was observed as spots in the internal peripheral region of the nucleus. As a quantitative indicator of the colocalization of PREBP-GFP with the nucleus, Pearson's R value was calculated for each individual parasite cell observed. The results for each stage of development are shown in Fig 2. During the ring stage, a negative correlation was observed between the nucleus and PREBP-GFP colocalization, with the highest correlation in trophozoites and a slightly lower correlation in schizonts. On the other hand, the positive correlation remained consistent throughout both the trophozoite and schizont stages. Furthermore, we conducted a quantitative image analysis to assess the colocalization of PREBP and nucleus at each stage of development (Fig 2). The results are presented in Fig 2, and the quantitative data support the observation that PREBP was predominantly localized outside of the nucleus during the ring stage, while in the trophozoite and schizont stages, it exhibited a tendency to colocalize with the nucleus. The stages in which PREBP-GFP localized at the nucleus matched the timing of the elevated expression of the target *pf1-cys prx* [16]. These results showed the possibility that the transcriptional enhancer activity may be regulated by the cellular localization of PREBP.

### Recruitment of PREBP on the *cis*-enhancer of the *pf1-cys-prx* gene on the chromosome in the living parasite

In a previous study [18], we showed specific interaction between PREBP and the DNA sequence of the *cis*-enhancer element of the *pf1-cys-prx* gene by an *in vitro* electrophoresis mobility shift assay. In the present study, to investigate the association of PREBP with the *cis*-

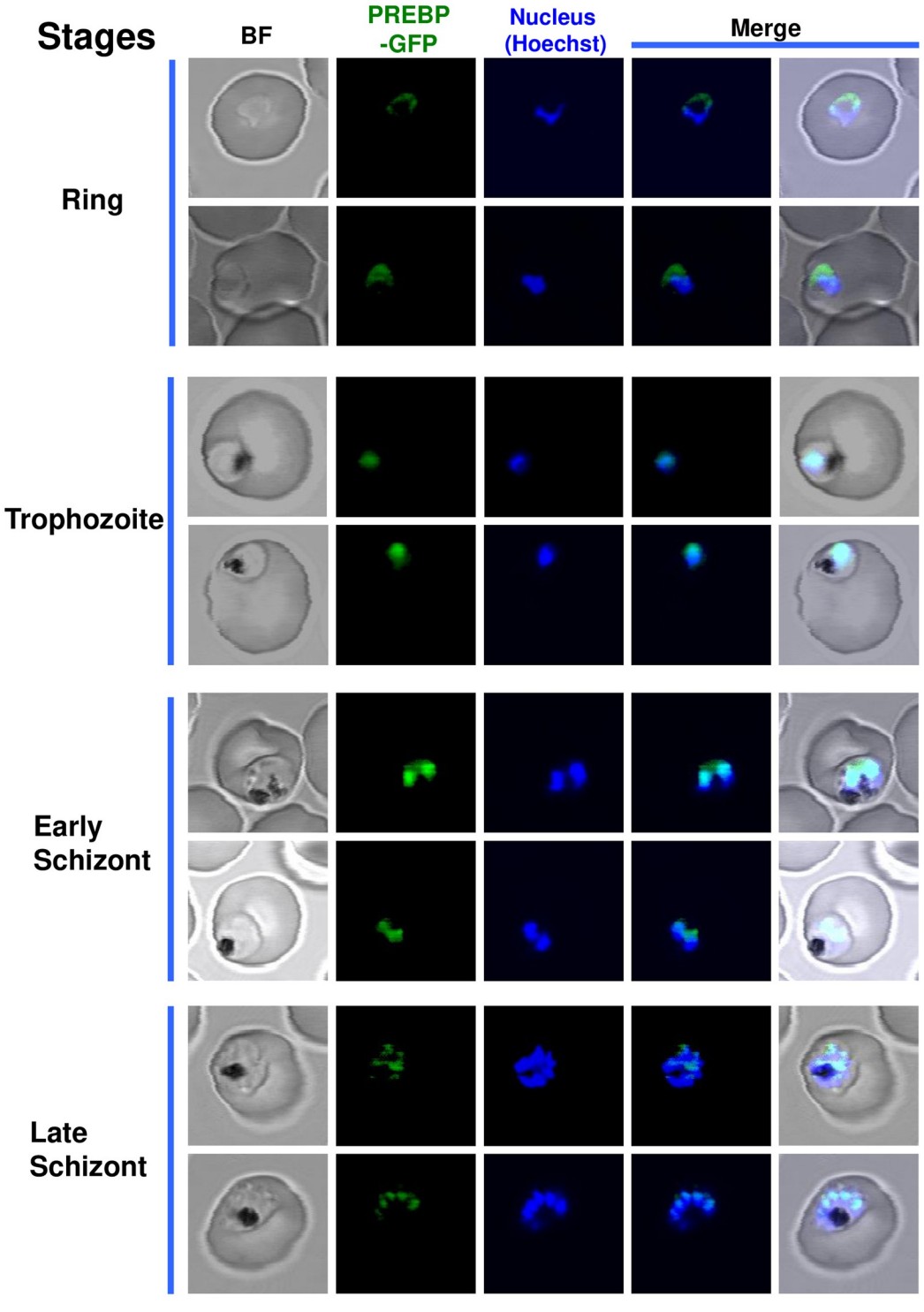

**Fig 1. Cellular localization of PREBP-GFP fusion protein in *P. falciparum* cells during the intraerythrocytic stage.**
Living parasites that expressed PREBP-GFP fusion protein were stained with Hoechst 33342 for visualization of the nucleus.
Ring, trophozoite, early schizont (with 2–3 numbers of the nucleus), and late schizont-stage parasites were observed under
407-nm emission for the detection of Hoechst (blue color), and under 488-nm emission for the detection of PREBP-GFP
(green color). "BF" indicates bright-field images. "Merge" indicates merged images of Hoechst and PREBP-GFP, or those of
Hoechst, PREBP-GFP, and BF. The scale bar is the same for all images.

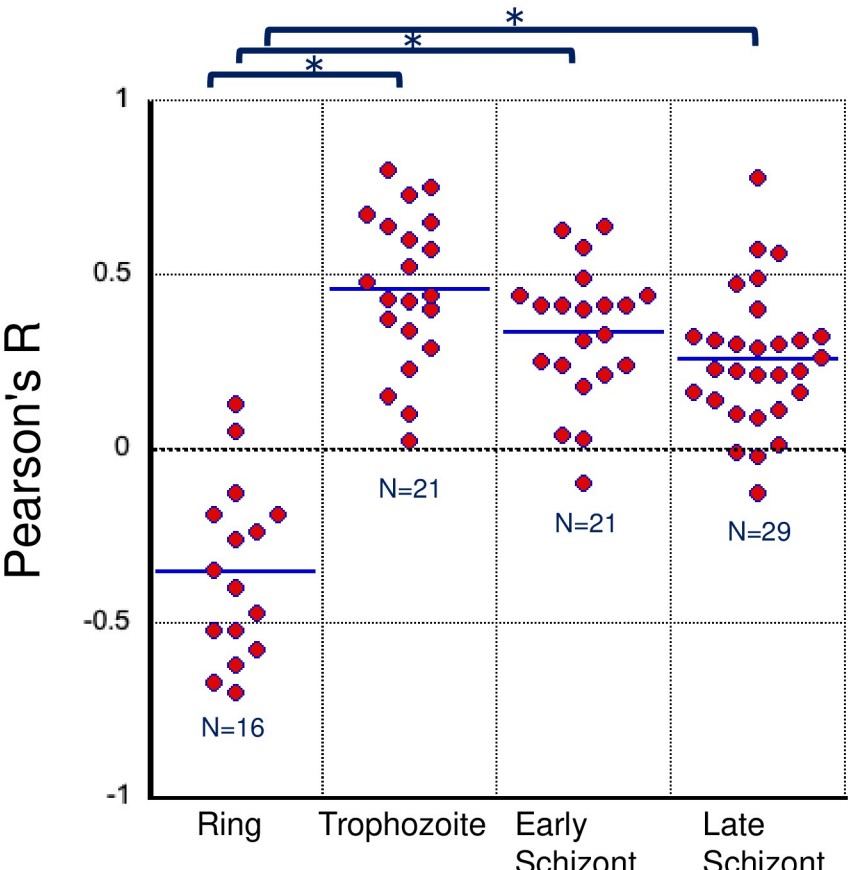

**Fig 2. Quantitative image analysis of cellular colocalization between the nucleus and PREBP-GFP at different stages of development.** Pearson's R value for the colocalization between the nucleus, stained with Hoechst (blue) and PREBP-GFP (green), was calculated. Raw data points are represented as dots on the graph, with the average indicated as a bar for each stage of development. The number of parasite cells analyzed in each stage of development is also displayed on the graph. Statistical significances (p<0.01) are indicated by asterisks.

enhancer regions of *pf1-cys-prx* on the chromosome in parasite cells, we performed ChIP assays with specific antibodies against the recombinant PREBP. Cultured parasites were tightly synchronized at the trophozoite/schizont stages at 28–32 h after invasion into erythrocytes. This timing was the same as that of the parasite stage, at which the expression of the *pf1-cys-prx* reached its peak [16]. The parasites were purified and treated with formaldehyde to cross-link protein-DNA complexes. The fragments from the 5′ region of the *pf1-cys-prx* and the ORF in immunoprecipitates were quantified by real-time PCR with sequence-specific primer sets. The amount of DNA precipitated with the anti-PREBP antibody was normalized to that of the input sample. As a result, the level of PREBP recruitment at the *cis*-enhancer of the *pf1-cys-prx* was higher than in other regions (Fig 3).

## Search for regions of PREBP responsible for its transcription enhancer activities

PREBP is a novel transcription factor and does not contain any characteristic domains, with the exception of putative RNA or single-stranded DNA binding KH domains. To search for functional domains of PREBP, the transcriptional enhancer activity of PREBP with regional

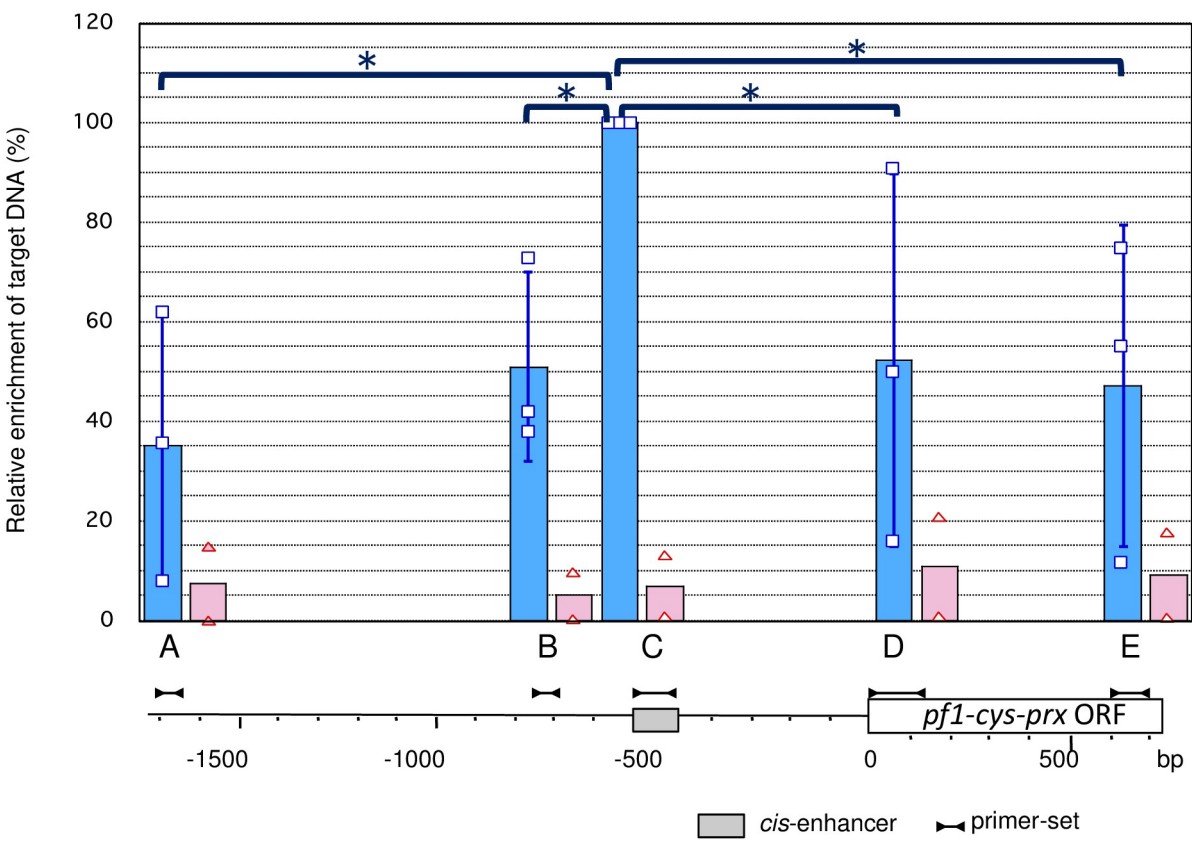

**Fig 3. Relative *pf1-cys-prx* promoter-PREBP associations at trophozoite/schizont stage.** ChIP assays were performed with chromatin from the parasite and the antibody against PREBP. Data shown were derived from quantitative real-time PCR with the primer pairs indicated at the bottom of the graph. The position of each primer set is indicated under the graph. The relative amount of immunoprecipitated DNA was normalized to the initial amount of DNA in the experiment. The highest average value (primer set C) was defined as 100% in each assay. In all cases, data represent the mean ± standard deviation of three independent assays (blue bars). The raw data points for each assay are indicated as squares. As a negative control, ChIP assays were also performed using normal mouse IgG, and data from two independent assays are indicated on the graph as triangles (raw data points) and red bars (average). Data sets obtained from Primer sets A, B, D, and E were compared with the data set of Primer set C and, as a result, statistical significance (p<0.01) was observed in all cases (indicated by asterisks).

deletion was measured via transient luciferase reporter assay systems. The principle of these systems is indicated in Fig 4A. The basal plasmid for the assay was p1-10R-PREBP, in which the firefly luciferase gene is located under the control of the 5′ promoter sequence of the *pf1-cys-prx* gene, and the PREBP overexpression cassette was located on the same plasmid [18]. A series of deletions were introduced to the ORF of PREBP in the p1-10R-PREBP plasmid. Then, if the deleted regions were essential for the activity, the expression of luciferase under the control of the promoter region of *pf1-cys-prx* would be significantly reduced. The structure of PREBP deletion mutants used for the assays is shown in Fig 4B. As we already showed in our previous report [16], with overexpression of the full-sequence of PREBP, the luciferase expression under the control of the 5′ promoter sequence of *pf1-cys-prx* was elevated more than 20-fold in comparison to the same plasmid without the PREBP sequence (p1-10R mock). The background luciferase activity observed upon introduction of the p1-10R mock was thought to be due to endogenous PREBP activity. The average expression level of luciferase with the full sequence of PREBP was defined as 100% of the activity of PREBP. At first, luciferase assays with a series of deletions of the N terminal region of the PREPB were performed. The plasmid constructs were introduced into parasites synchronized in the ring stage

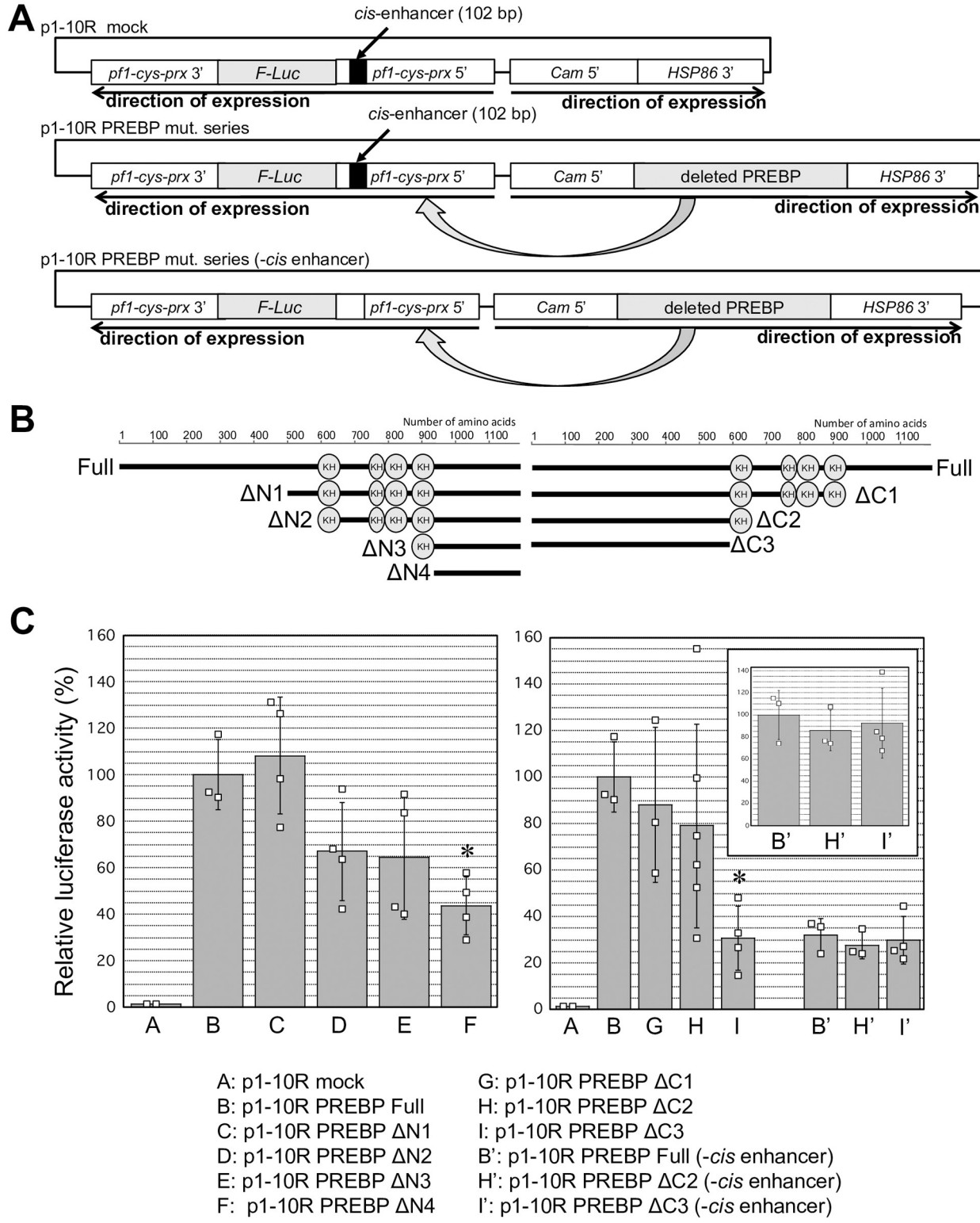

**Fig 4. Activities of PREBP deletion mutants.** (A) Schematic diagram of plasmid constructs used for the transient luciferase assay in *P. falciparum*. The overexpressed PREBP mutants under the control of the *Cam* 5′ sequence could associate with the *pf1-cys-prx* promoter, which was located on the same plasmid and which controls the expression of firefly-luciferase. (B) The structures of the deletion mutants used for the luciferase assays. (C) The dual luciferase assay. The plasmids shown in (A) with various mutants shown in (B) were transfected into the parasite synchronized in the ring stage. For the dual luciferase assay, the pHC1-Rluc plasmid, which expresses Renilla luciferase, was co-transfected in each assay. After 20–24 h, the luciferase

activities of the transfected parasites were measured, and the ratio of firefly/Renilla luciferase activity was calculated. The average activity of full-sequence PREBP was defined as 100% in each assay. The data are indicated as the mean values of three independent assays with SD. Asterisks indicate a significant difference ($p < 0.01$) in comparison to the activity of full-sequence PREBP. The left panel shows the activity of the PREBP deletion series from the N terminus, and the right panel shows that of the deletion series from the C terminus. The inset shows the activity of the PREBP deletion series from the C-terminus against promoters lacking the 102-bp *cis* enhancer. In the inset, the average activity of the full-length PREBP is shown as 100%. The raw data points for each assay are indicated as squares.

by electroporation. After 20–24 h, the luciferase activity in the trophozoite-schizont parasites was determined. As a result, the PREBP activity tended to decrease according to the absence of its KH domains, and the activity was significantly reduced when it lacked all hypothetical KH domains ($p < 0.01$) (Fig 4C, left panel). When up to three KH domains were deleted, there was a trend toward decreased activity, but the decrease was not statistically significant. Then, luciferase assays with a series of deletions of the C terminal region of PREPB were also performed. As a result, PREBP activity was not significantly reduced until it lacked 3 of 4 KH domains. Then, the activity was significantly decreased when it lacked all the hypothetical KH domains ($p < 0.01$) (Fig 4C, right panel). Both deletion series of N- and C- terminal regions indicated that the region including 4 repeated hypothetical KH domains is responsible for the activity of PREBP.

Furthermore, to gain insight into the contribution of the KH domains to the recognition of the *cis*-DNA sequence, we analyzed the effect of deletion of the 102-bp *cis*-enhancer from the *pf1-cys-prx* promoter sequence (Fig 4A). In a previous study, it was shown that without the 102-bp sequence of the *cis*-enhancer region in the promoter sequence, the transcription enhancer activity of PREBP was reduced to approximately 30% of that of the full-promoter sequence [18]. This result suggested that the activity of PREBP depended on the specific sequence of the *cis*-enhancer in the promoter. Then, in the present study, we analyzed the effects of a series of C-terminal deletions of PREBP on its activity with the promoter sequence lacking the *cis*-enhancer sequence. As a result, the activity of PREBP on the promoter sequence without the *cis*-enhancer was not affected by the absence of all KH domains (Fig 4C, right panel). These results comprehensively suggested that the region containing the hypothetical KH domains of PREBP was responsible for the activity of PREBP, which depended on the recognition of the *cis*-DNA sequence.

## Searching for hypothetical transcription factors with KH-domains in Apicomplexan organisms

Our results showed that a region of PREBP, which contained 4 repeated hypothetical KH domains, was responsible for the specific recognition of the *cis*-enhancer and its transcriptional enhancer activity. These results led us to hypothesize that in *Plasmodium* or other Apicomplexan organisms, KH domain-containing proteins might act as transcription factors. Then, to find candidates for the hypothetical transcription factors with KH domains in Apicomplexan organisms, we first surveyed the well-organized Apicomplexan genome databases, PlasmoDB [32], ToxoDB [33], and CryptoDB [34]. We screened the genes encoding proteins of the three representative parasite species, *P. falciparum*, *Toxoplasma gondii*, and *Cryptosporidium parvum* from each database, with hypothetical KH domains using the term "KH". Then, we removed several proteins that are orthologs of well-known ribosomal RNA-related proteins from the protein lists indicated as results of the search (S2 Table). Next, the secondary structures of these candidates were predicted by the SMART program [35]. At this step, factors that were not predicted to contain KH domains were excluded (S2 Table). Finally, we found 4 proteins from the *P. falciparum* genome, 10 proteins from the *T. gondii* genome, and 6

proteins from the *C. parvum* genome (Table 1). The secondary structure prediction indicated that each candidate protein might contain 1–4 repeats of hypothetical KH domains (Fig 5). Then, the amino acid sequences of these candidate proteins were subjected to the ortholog group search program BLASTClust [36]. For the analysis, we added the sequence of the FUBP1, a well-characterized human transcription factor with KH domains [22–25] to the candidate group. As a result, the candidate proteins were classified into 3 putative ortholog groups (Fig 5). The biggest hypothetical group (Group 1, Fig 5) contained human FUBP1, PREBP, and also 3 other *Plasmodium* proteins, 3 *T. gondii* proteins, and 3 *C. parvum* proteins. Another group (Group 2, Fig 5) contained 3 proteins such as *P. falciparum*, *T. gondii*, and *C. parvum* proteins. These 3 proteins in Group 2 have quite similar structures and also have the splicing factor 1 helix-hairpin (SF1H-H) domain; thus, they could be considered as conserved splicing factors. The other small group (Group 3, Fig 5) consisted of 2 proteins from *T. gondii* and *C. parvum*. Six individual proteins were defined as orphans. The putative orthologs in Group 1, which contains the already known transcription factors human FUBP1 and PREBP, could be considered as promising candidates for a novel transcription factor superfamily with KH domains. Three hypothetical proteins (PF3D7_0302800, PF3D7_0605100, and PF3D7_1415300) of *P. falciparum* were included in this group.

To corroborate the similarity relationships between candidate proteins, Apicomplexan genome databases were screened with each candidate protein via the BLASTP program.

**Table 1. List of candidates for the KH domain, including proteins found from Apicomplexan genome databases.**

| Gene name | Annotation | Protein length | GenBank Accession No. |
|---|---|---|---|
| *Plasmodium falciparum* | | | |
| PREBP (PF3D7_1011800) | PRE-binding protein | 1139 aa | XM_001347364 |
| PF3D7_0302800 | Conserved Plasmodium protein, unknown function | 419 aa | XM_001351067 |
| PF3D7_0623600 | Transcription or splicing factor-like protein, putative | 615 aa | XM_961125 |
| PF3D7_0605100 | RNA-binding protein, putative | 755 aa | XM_960948 |
| PF3D7_1415300 | RNA-binding protein Nova-1, putative | 337 aa | XM_001348288 |
| *Toxoplasma gondii* | | | |
| TGGT1_209210 | Hypothetical protein | 1492 aa | XM_018779456 |
| TGGT1_212980 | Hypothetical protein | 715 aa | XM_002371006 |
| TGGT1_216670 | FUSE-binding protein 2 / KH-type splicing regulatory protein | 941 aa | XM_002370934 |
| TGGT1_217880 | Putative RNA-binding protein Nova-1 | 412 aa | XM_002371314 |
| TGGT1_235930 | Domain K- type RNA binding proteins family protein | 570 aa | XM_002368937 |
| TGGT1_237550 | Hypothetical protein | 1489 aa | XM_018780499 |
| TGGT1_241170 | Hypothetical protein | 1275 aa | XM_002366685 |
| TGGT1_271250 | Hypothetical protein | 418 aa | XM_002365750 |
| TGGT1_320080 | Hypothetical protein | 1673 aa | XM_018782983 |
| TGGT1_314860 | Zinc knuckle domain-containing protein | 723 aa | XM_002364635 |
| *Cryptosporidium parvum* | | | |
| cgd4_130 | RRM domain and KH domain protein (SPAC30D11.14-like KH) | 801 aa | XM_625622 |
| cgd7_720 | PASILLA splice variant 3-like 2KH domains, transmembrane domain at C-terminus | 364 aa | XM_628259 |
| cgd4_1210 | Ms15p | 471 aa | XM_625718 |
| cgd1_1280 | Domain KOG1676, K-homology type RNA binding proteins | 460 aa | XM_627972 |
| cgd7_1890 | KH domain protein | 818 aa | XM_628364 |
| cgd2_2940 | Conserved hypothetical protein | 1132 aa | XM_626491 |

Gene names are the ID in PlasmoDB [30] release 58, 2022 (for *P. falciparum*), ToxoDB [31], release 58, 2022 (for *T. gondii*), and CryptoDB [32] release 58, 2022 (for *C. parvum*).

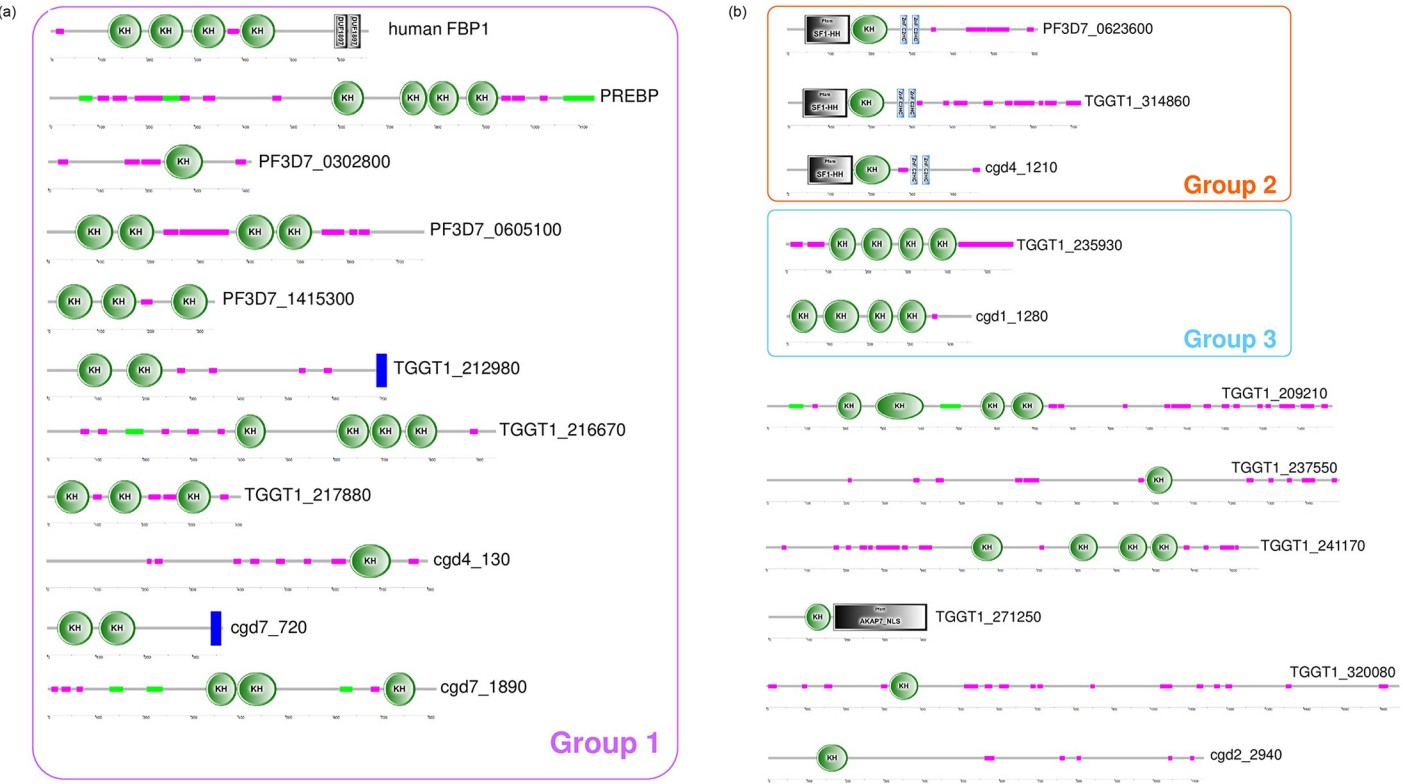

**Fig 5. Clustering of Apicomplexan KH domain-containing proteins into putative ortholog groups.** Three putative ortholog groups predicted by the BLASTClust analysis are indicated. Each candidate protein is illustrated with its secondary structure predicted by the SMART program. Green circles indicate KH domains. Pink lines indicate low-complexity regions detected in the SEG program. Green lines indicate coiled-coil regions. Blue boxes indicate transmembrane domains. The "SF1H-H" domain is a splicing factor 1 helix-hairpin domain. The DUF_1897 domain, Zinc-finger domain (ZnFC2HC), and AKAP7_NLS domain are indicated with square boxes with each name.

Among the putative homologs identified by screening with candidate proteins of Group 1 (Fig 5), 70% were confirmed as actual members of Group 1 (S3 Fig). Similarly, during the search for putative homologs by screening proteins from Group 2 (Fig 5), it was discovered that 85.7% indeed belonged to Group 2, demonstrating a strong similarity (S3 Fig). Thus, in relation to putative orthologs within Group 1 and Group 2, the mutual BLASTP search provided further support for the results obtained from BLASTClust. Proteins in Group 3 have no putative homolog in the *P. falciparum* genome. Proteins classified as orphan proteins tended to have no homologs in the Apicomplexan genomes, even though some of them showed similarity to proteins classified as group 1 or group 3.

### Cellular localization of newly identified factors through intraerythrocytic development of the parasite

We analyzed the cellular localization of the newly identified factors of *P. falciparum* that belong to the putative ortholog group 1. Parasite lines that express each of the PF3D7_0302800, PF3D7_0605100, and PF3D7_1415300 proteins, which were fused with the GFP, were developed according to the same strategy that was used to make the parasite line that expressed PREBP-GFP fusion protein (S1 Fig). The PF3D7_0605100-GFP-expressing transgenic parasite line could not be established because stable integration of the GFP gene

into the PF3D7_0605100 locus could not be confirmed (S4 Fig). We observed the cellular localization of PF3D7_0302800-GFP and PF3D7_1415300-GFP using confocal laser microscopy. The localization of PF3D7_0302800-GFP matched the nuclear location in the trophozoite stage (Fig 6). As a quantitative measure of the colocalization of PF3D7_0302800-GFP with the nucleus, Pearson's R value was calculated for each individual observed parasite cell. The results indicate that during the trophozoite stage, the R values are significantly higher than those observed in other stages (Fig 7). An analysis of colocalization between PF3D7_1415300-GFP and the nucleus was conducted; however, no consistent pattern of colocalization was observed (S5 and S6 Figs). These results suggested that PF3D7_0302800 could possibly function as a transcription factor in the nucleus of trophozoites.

## Discussion

We showed that PREBP was localized at the parasite nucleus and was associated with its target *cis*-enhancer sequence in the trophozoite/schizont stages, in which PREBP acted as a transcriptional activator. The observations of PREBP-GFP localization were consistent with a previous report [18] in which PREBP was detected in the trophozoite/schizont nuclear fraction by immunoprecipitation and Western blot analysis. The region, with 4 repeated hypothetical KH domains of PREBP, was responsible for the specific transcriptional regulatory activity depending on the target *cis*-enhancer. We then searched other candidate proteins for unknown transcription factors with KH domains in the genome of Apicomplexan parasites. Several Apicomplexan factors with KH domains were detected in the same putative ortholog group, which included PREBP. One of the *Plasmodium* factors that belonged to the putative ortholog group was found to localize at the parasite nucleus, suggesting that this factor might be a novel transcription factor of *Plasmodium*.

Typically, transcription factors are structurally composed of the DNA binding domain and trans-activation domain. Trans-activation domains are responsible for binding to other transcription co-regulator proteins, such as components of the basal transcription machinery or chromatin remodeling factors [37]. In contrast, DNA binding domains contain binding sites for specific double-stranded-DNA sequences of *cis*-regulatory elements. Leucine-zipper, helix-turn-helix, and zinc fingers are known examples of DNA binding domains of transcription factors. PREBP does not contain any typical domains that have been found in other transcription factors, and only KH domains are predicted from the amino acid sequence of PREBP. The KH domain is known as an RNA- or single-stranded DNA-binding domain and is not expected to show specific binding to actual double-stranded DNA. However, in this study, the results of the assay with PREBP deletion mutants indicated that the region with KH domains was responsible for specific *cis*- enhancer recognition and its binding.

In humans, transcription factors, termed FUBP1, which contain KH domains for specific binding to a single-stranded *cis*-element have been reported [22–25]. FUBP1 recognizes FUSE, the *cis*-element of the *c-myc* gene, and regulates its transcription. The detailed molecular mechanism of human FUBP1 was investigated, and it is known that FUBP1 binds to FUSE, which dissociates into single-stranded DNA through the functions of chromatin remodeling factors according to the transcriptional activation [24]. In our previous study, we detected *in vitro* binding of PREBP to the 102-bp *cis*-enhancer of *pf1-cys-prx* by electrophoresis mobility shift assays. In these assays, double-stranded DNA was used as a labeled probe, and shift bands corresponding to the DNA probe-PREBP complex were detected. There was a possibility that the probes were partially melted into single-stranded DNAs and that PREBP might bind specifically to such single-stranded DNA probes.

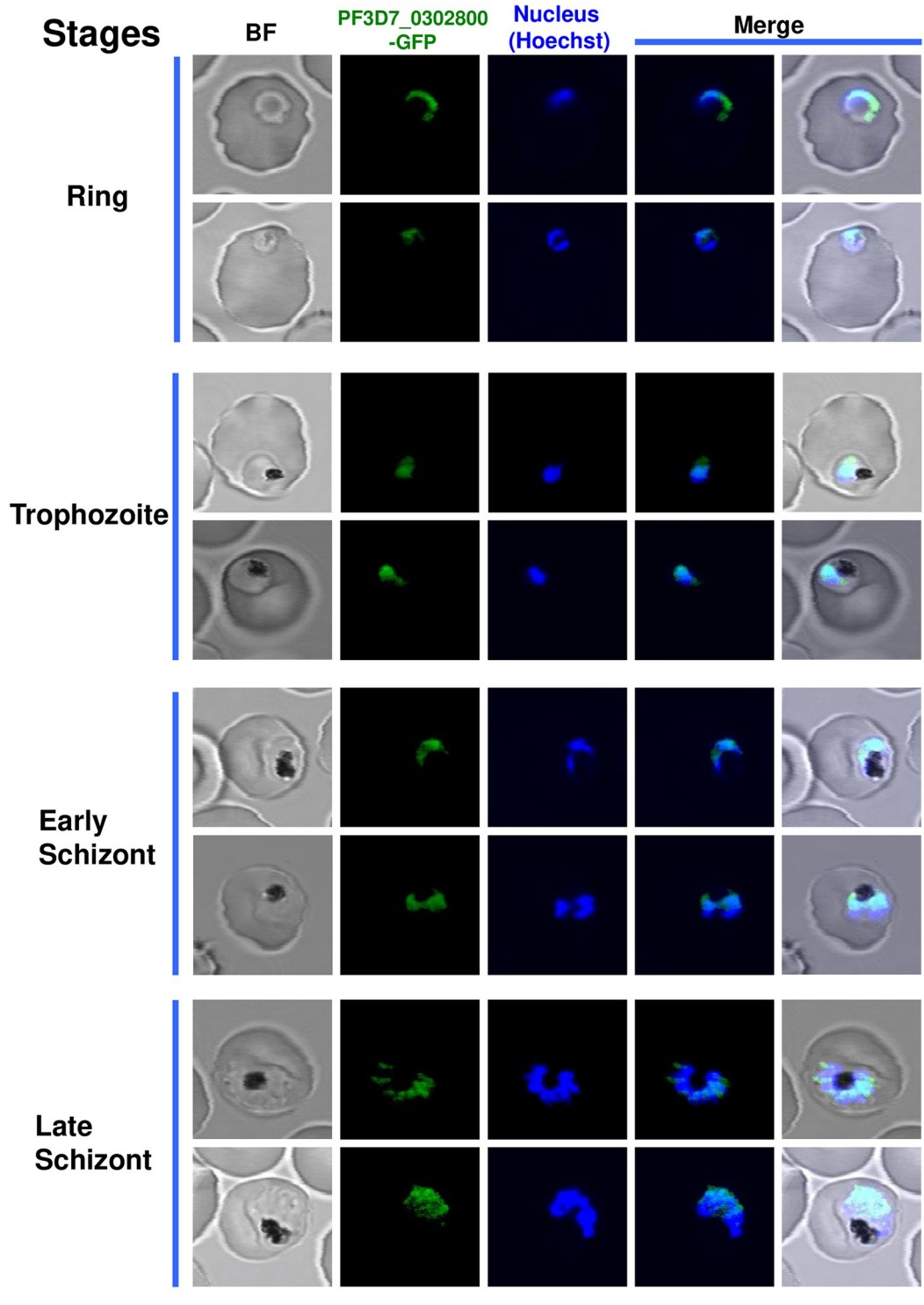

**Fig 6. Cellular localization of the PF3D7_0302800-GFP fusion protein in *P. falciparum* cells during the intraerythrocytic stage.** Living parasites that expressed the PF3D7_0302800-GFP fusion protein were observed in the same manner used for observing the expression of the PREBP-GFP. Parasites of each developmental stage were observed under 407-nm emission for the detection of the Hoechst (blue color), and under 488-nm emission for the detection of PF3D7_0302800-GFP (green color). "BF" indicates bright-field images. "Merge" indicates merged images of Hoechst and PREBP-GFP or those of Hoechst, PF3D7_0302800-GFP, and BF. The scale bar is the same for all images.

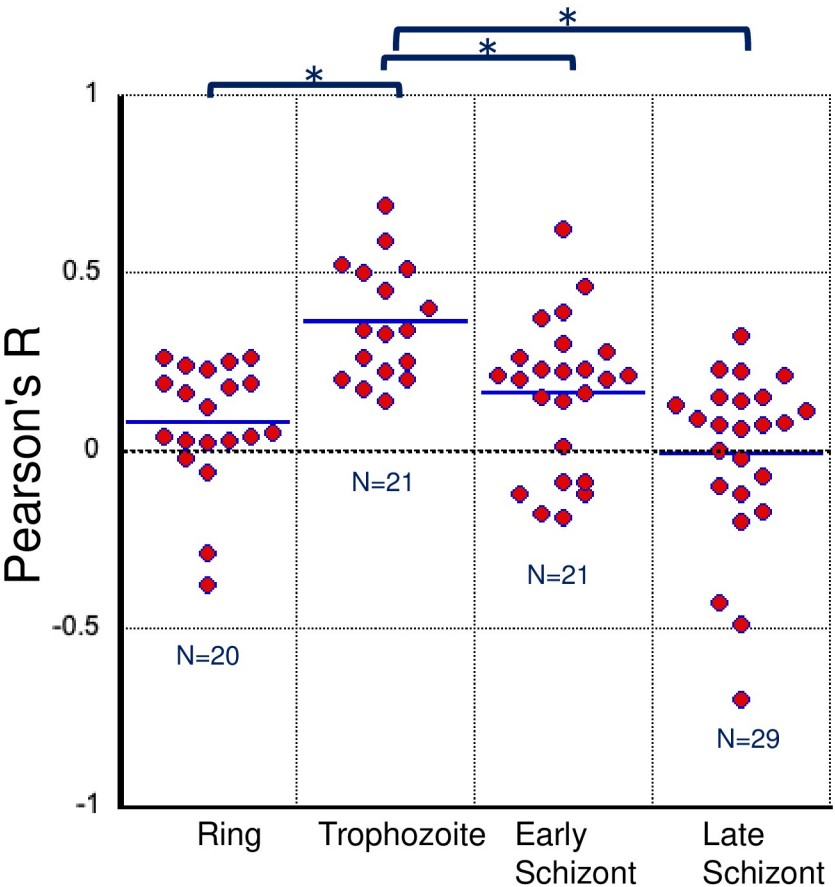

**Fig 7. Quantitative image analysis of cellular colocalization between the nucleus and PF3D7_0302800-GFP at different stages of development.** Pearson's R value for colocalization between the nucleus, stained with Hoechst (blue) and PF3D7_0302800-GFP (green), was calculated. Raw data points are represented as dots on the graph, with the average indicated as a bar for each stage of development. The number of parasite cells analyzed in each stage of development is also shown in the graph. Statistically significant differences ($p < 0.01$) are indicated by asterisks.

Our previous study [17] suggested that the 102-bp *cis*-enhancer of *pf1-cys-prx* might be the target for histone acetylation and site-specific recruitment of *P. falciparum* (Pf) GCN5, a histone acetyl-transferase [38], at the same timing as transcriptional activation. In this study, the recruitment of PREBP to the same *cis*-enhancer was also detected by the ChIP assay. To date, no information on the transactivation domains of PREBP has been reported; however, PREBP can likely interact with PfGCN5, chromatin remodeling factors, and/or basal transcription machinery, via its unknown transactivation domain. PREBP has 4 KH domains, and although removal of up to 3 of them did not result in a significant decrease in its activity, it did cause PREBP activity to become unstable. The presence of the 4 KH domains may be responsible for the stable recognition of the 102-bp *cis*-enhancer, which will be further elucidated in future studies. Even when all of the KH domains were removed, PREBP still s approximately 30% activity independent of the 102-bp *cis*-enhancer. This residual activity could be due to nonspecific binding to promoter DNA and interaction with transcriptional cofactors such as PfGCN5.

The bioinformatics assay in this study suggested some candidates for the unknown transcription factor with KH domains in Apicomplexan parasites. From the *P. falciparum* genome,

3 more candidates were suggested in addition to PREBP. All of these putative factors (PF3D7_0302800, PF3D7_0605100, and PF3D7_1415300) are well conserved among the other *Plasmodium* species, *P. berghei*, *P. chabaudi*, *P. cynomolgi*, *P. knowlesi*, *P. reichenowi*, *P. vivax*, and *P. yoelii* (S2 Table). Currently, PF3D7_0302800 is annotated as "RNA-binding protein, putative" in PlasmoDB (release 65), and contains one hypothetical KH domain. In the present study, the cellular localization of PF3D7_0302800 was analyzed, and it was found to be localized in the parasite nucleus during the trophozoite stage. Although it is not possible to conclude that PF3D7_0302800 is a transcription factor without further detailed analysis, it is possible that it functions as a transcription factor in the nucleus and regulates the gene expression at these stages. PF3D7_0302800 had only one KH domain, and if it is indeed a transcription factor, how such protein structure is involved in recognition of the target *cis*-element needs to be clarified by future studies. PF3D7_0605100 is annotated as "RNA-binding protein, putative", and it contains 4 hypothetical KH domains. A transgenic parasite line that expresses PF3D7_0605100 and GFP fusion protein could not be established in the present study. PF3D7_0605100 might be a novel transcription factor, and its essential function may be interrupted by fusion to the GFP protein. PF3D7_1415300 is annotated as "RNA-binding protein Nova-1, putative" and contains 3 hypothetical KH domains. Nova-1 is a human neuronal RNA binding splicing factor with KH domains [39, 40] and PF3D7_1415300 also has similarity to Nova-1; thus, this factor was annotated as a putative Nova-1 homolog. PF3D7_1415300 was also considered as a putative splicing factor [41]. The cellular localization of PF3D7_1415300 was detected as being the outside of the nucleus in the present study, suggesting that the factor had a function other than that of being a transcription factor. If PF3D7_1415300 was a splicing factor, the predicted localization would be nuclear. Thus, PF3D7_1415300 may not function as a splicing factor, either. Both transcription factors and non-transcription factors were likely to be included in the same putative ortholog group.

In considering the evolutionary origins of these KH proteins, we focused on free-living organisms that share a common ancestor with Apicomplexa. The whole-genome sequences of *Chromera velia* and *Vitrella brassicaformis* were previously reported [42]. These 2 species belong to the Phylum Chromerida, which are free-living photosynthetic algae that are philologically closely related to parasitic Apicomplexan organisms. Interestingly, the number of genes that encode DNA- or RNA-binding proteins was drastically reduced in Apicomplexan parasites in comparison to the Chromerida species. Such genes might be lost from the genome of ancestor alga of Apicomplexan parasites when they obtained an intracellular parasitic lifestyle. The number of proteins with KH domains was also reduced in the Apicomplexan genome. Apicomplexan parasitic protozoa might utilize a small number of proteins with KH domains for the regulation of RNA metabolism and transcription. In addition, recent studies have shown that various RNA-binding proteins act on chromatin in mammalian cells, interacting with gene promoters and transcription factors to activate transcription [43]. RNA-binding proteins have a broader range of functions than were expected and may be involved in transcriptional regulation by interacting directly with transcription factors on chromatin. In Apicomplexa, only a few typical transcription factors have been found, multiple functions may be concentrated in a few nucleic acid-binding proteins, and future studies are needed to determine their evolutionary significance. Because of the limited number of these KH factors present in the parasite, they are expected to affect the transcription of multiple genes. In the future, it will be necessary to determine which gene expression is regulated by these factors.

To date, despite extensive evidence supporting AP2s as transcriptional regulators in *Plasmodium* species, the detailed mechanisms of transcriptional regulation in *Plasmodium* and other Apicomplexan organisms are still not fully understood. The discovery of PREBP and putative transcription factors with KH domains suggests the possibility that multi-functional

proteins with KH domains have evolved in Apicomplexan organisms and regulate transcription. Further detailed characterization of the functional mechanisms of these putative transcription factors of malaria parasites may help to understand the evolutionary process of specific regulation for DNA and RNA metabolism in Apicomplexan parasites and could lead to the identification of unique drug targets for malaria.

## Supporting information

**S1 Fig. Strategies for making transgenic parasite lines that express each putative transcription factor and GFP fusion protein.** (A) The strategy for the construction of plasmid vectors used for the establishment of transgenic parasites. The partial sequence of transcription factor (TF) and the GFP sequence were amplified by PCR. The primers used for the PCR reactions were designed so that the amplicon retained *Xho*I and *Not*I sites at both ends. The PCR products were digested by *Xho*I and *Not*I and then cloned into the *Xho*I site of the pHC1 vector. The vector contains the *T. gondii* DHFR-TS gene as a resistant gene for selection by pyrimethamine. (B) Schematic illustration of the transcription factor gene (PREBP or PF3D7_032800) locus and organization following the single-crossover homologous recombination event. After recombination, TF and GFP fusion proteins were expressed under the control of the endogenous 5′ promoter. (C) PCR to check for genomic integration. Genomic DNA was extracted from the parent and each of the recombinant clonal parasite strains, and PCR was performed using them as templates. Primer sets were used to detect the ORF of each target factor gene (primer set G) and the ORF formed by the fusion of each factor gene and the GFP gene (primer set R). The primer locations are shown in the schematic illustrated in S1B Fig. The primer sequences are shown in S4B Fig. (D) Observation of the recombinant parasite strains using fluorescence microscopy. Each living parental or recombinant parasite strain was observed under a fluorescent microscope in bright (BF) and dark (FL) field. Parasites visible in the BF are indicated by black arrows, and parasites emitting GFP fluorescence visible in the DF are indicated by yellow arrows. Scale bar: 20 μm.
(PDF)

**S2 Fig. Western blot analysis of ChIP products.** The chromatonin immunoprecipitation (ChIP) assay was conducted according to the methodology outlined in the Materials and Methods section. During the immunoprecipitation step, we used a rabbit polyclonal antibody against recombinant-PREBP [18] or an equivalent amount of normal rabbit IgG for the immunoprecipitation of the chromatin sample. Subsequently, 20 μL of Dynabeads eluates containing the antigen-antibody complex were mixed with 5 μL of 5X SDS-PAGE sample buffer (comprising 5% SDS, 0.1% Bromophenol Blue, 312.5 mM Tris-HCl at pH 6.8, and 50% glycerol), and 25 mM DTT was added, followed by incubation at 100˚C for 10 minutes. The mixture was then subjected to 5–20% SDS–PAGE. Following electrophoresis, the proteins were electrophoretically transferred to polyvinylidene difluoride sheets (Immobilon; Merck-Millipore) and probed with the anti-PREBP antibody. Immune complexes were visualized using TrueBlot Anti-IgG HRP® antibody, Rabbit (Rockland Immunochemicals, Inc., Pottstown, USA), and SuperSignal™ West Dura Extended Duration Substrate (Thermo Fisher Scientific). Lane 1 corresponds to the precipitate with anti-PREBP, while Lane 2 corresponds to the precipitate with normal rabbit IgG. Molecular weight markers in kDa are indicated on the left for reference. The bands corresponding to PREBP are indicated with red arrows.
(PDF)

**S3 Fig. Mutual homology search of Apicomplexan genome databases with each candidate protein.** (A)The genome database for *P. falciparum* (Pf), *T. gondii* (Tg), and *C. parvum* (Cp)

was searched using the BLASTP program with each candidate protein. The proteins used for each BLASTP search are indicated at the top of each table. The genome databases used for each search are shown on the left side of each table, and homologs suggested by each BLASTP search are indicated inside tables with e-values. Putative homologs with e-values of $<1 \times 10^3$ are indicated. The protein itself, which was subjected to the search and indicated as a homolog, is not included in the table. Protein belonging to group 1 (Fig 5) is indicated with a purple box. Proteins indicated with yellow, light blue, and green boxes belong to group2, group 3, and orphans, respectively. "None" means that no homolog with a significant e-value was found. Proteins indicated with a light-green box are proteins without hypothetical KH domains. (B) The pie charts depict the distribution of homolog types identified through BLASTP searches for each candidate protein. The percentages representing each homolog type in the graph were calculated using the data presented in (A). Different homolog categories, including those belonging to group 1, group 2, group 3, orphans, and others (proteins not included in candidate proteins with KH domains), are represented by distinct colors: purple, yellow, light blue, green, and light green, respectively.
(PDF)

**S4 Fig. Attempted establishment of transgenic parasite line expressing PF3D7_0605100 and GFP fusion protein stably.** In order to express the fusion protein of PF3D7_0605100 and GFP for *P. falciparum*, the plasmid shown in S1A Fig was introduced into the cultured parasites via electroporation. Screening of genetically modified parasites was performed through cultivation in the presence of 0.1 μM pyrimethamine. Under these conditions, for the three strains introduced the plasmids for the fusion proteins of PREBP, PF3D7_0302800, and PF3D7_1415300 along with GFP, it took less than three months to observe the disappearance of pyrimethamine-induced parasite death, with an increase in growth speed. This suggested that the introduced plasmids containing drug resistance genes integrated into the genome. However, for the plasmid for the PF3D7_0605100-GFP fusion gene introduced parasites, pyrimethamine-induced parasite death was still observed even after three months from the transfection, indicating that the drug resistance gene did not integrate into the genome and remained episomal. At this point, a limiting dilution was performed with the goal of cloning all four recombinant parasites. For the PREBP, PF3D7_0302800, and PF3D7_1415300, stable parasite lines expressing the fusion proteins with GFP were successfully established (S1 Fig). However, for PF3D7_0605100, pyrimethamine-induced parasite death continued even after limiting dilution. Subsequently, RNA was extracted from all strains, and cDNA was synthesized for each, following the methods described in a previous report [16]. RT-PCR was conducted to confirm the expression of the respective genes fused with GFP. (A) PCR to confirm the expression of fusion gene. PCR was performed using cDNA prepared from RNA extracted from each parasite strain as a template. Primer sets were used to detect the cDNA of each target gene (primer set E) and the cDNA transcribed from the fusion mRNA of each factor gene and the GFP gene (primer set F). The primer sets and locations are the same as those shown in the primer sets G and R in S1B Fig. For PREBP, PF3D7_0302800, and PF3D7_1415300, clear PCR amplicons were observed using both primer sets E and F, indicating the expression of mRNA encoding the fusion of the GFP gene with each respective factor. In contrast, for PF3D7_0605100, only a faint band corresponding to the fusion gene was detected with primer set F, in comparison to the amplicons obtained with primer set E, suggesting that the expression of the fusion gene was minimal. These results led to the determination that a stable parasite strain expressing the PF3D7_0605100-GFP fusion gene could not be obtained. (B) Primer sequences used for RT-PCR.
(PDF)

**S5 Fig. Cellular localization of the PF3D7_1415300-GFP fusion protein in *P. falciparum* cells during the intraerythrocytic stage.** Living parasites that expressed the PF3D7_1415300-GFP fusion protein were observed by the same method that was used to observe the expression of PREBP-GFP. Parasites of each developmental stage were observed under 407-nm emission for the detection of Hoechst (blue) and under 488-nm emission for the detection of PF3D7_0302800-GFP (green). "BF" indicates bright-field images. "Merge" indicates merged images of Hoechst and PREBP-GFP or those of Hoechst, PF3D7_0302800-GFP, and BF. The scale bar is the same for all images.
(PDF)

**S6 Fig. Quantitative image analysis of cellular colocalization between the nucleus and PF3D7_1415300-GFP at different stages of development.** Pearson's R value for colocalization between the nucleus, stained with Hoechst, and PF3D7_1415300-GFP was calculated. Raw data points are represented as dots on the graph, with the average indicated as a bar for each developmental stage. For all observed parasite cells in the ring stage, no GFP-derived green fluorescence could be detected, rendering the calculation of the R value impossible. The number of parasite cells analyzed in each stage of development is also displayed in the graph.
(PDF)

**S1 Table. List of primers used for constructing the plasmids employed in this study.**
(DOCX)

**S2 Table. Candidate proteins with KH domains identified in searches of PlasmoDB, ToxoDB, and CryptoDB for the bioinformatic analysis.**
(DOCX)

**S3 Table. List of homologs of Group 1 proteins preserved in *Plasmodium* species.** Identity (%), Positives (%), Gaps (%) and E-value indicate alignment results through BLAST when comparing each homolog with the query protein.
(DOCX)

## Acknowledgments

We are grateful to Dr. Keiko Takemoto of Kyoto University (Kyoto, Japan) and Dr. Takeshi Yasuda of the National Institute of Radiological Sciences (Chiba, Japan) for their helpful and critical suggestions and comments on this work.

## Author Contributions

**Investigation:** Kanako Komaki-Yasuda.

**Methodology:** Kanako Komaki-Yasuda.

**Project administration:** Kanako Komaki-Yasuda, Shigeyuki Kano.

**Writing – original draft:** Kanako Komaki-Yasuda.

**Writing – review & editing:** Shigeyuki Kano.

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
