## [Decision Letter · Decision Letter 0]

13 Feb 2023

PONE-D-23-00650The RNA-binding KH-domain in the unique transcription factor of the malaria parasite is responsible for its transcriptional regulatory activityPLOS ONE

Dear Dr. Komaki-Yasuda,

Thank you for submitting your manuscript to PLOS ONE. After careful consideration, we feel that it has merit but does not fully meet PLOS ONE’s publication criteria as it currently stands. Therefore, we invite you to submit a revised version of the manuscript that addresses the points raised during the review process.

You'll see that referee #1 only raised minor issues, whereas referee #2 raised three points regarding Figures 1 & 3 and clarification of a section of the Methods. None of the issues raised seem too difficult to rectify, and to give you the chance to improve Figures 1 & 3 I have marked your submission down for major revision.

We look forward to receiving your revised manuscript.

Kind regards,

Gordon Langsley

Academic Editor

PLOS ONE

Journal Requirements:

“This work was 596 supported by JSPS KAKENHI Grant Numbers 17K08817 and 20K07473.”

“This work was supported by JSPS (Japan Society for the Promotion of Science) KAKENHI Grant Numbers 17K08817 and 20K07473 (to KKY).

The funder’s URL: https://www.jsps.go.jp/

Reviewers' comments:

Reviewer's Responses to Questions

**Comments to the Author**

1. Is the manuscript technically sound, and do the data support the conclusions?

Reviewer #1: Yes

Reviewer #2: Partly

2. Has the statistical analysis been performed appropriately and rigorously? 

Reviewer #1: N/A

Reviewer #2: No

3. Have the authors made all data underlying the findings in their manuscript fully available?

Reviewer #1: Yes

Reviewer #2: No

4. Is the manuscript presented in an intelligible fashion and written in standard English?

Reviewer #1: Yes

Reviewer #2: Yes

5. Review Comments to the Author

Reviewer #1: The authors present a well done work on the identification of functional markers of a transcription factor that they had previously described (PREBP). This protein has 4 KH domains known to bind to either RNA or single stranded DNA. By working on plasmids expressing a luciferase activity regulated by PREBP, the authors show that the presence of these domains promotes the efficiency of expression and that they are therefore important in the regulation of transcription. Based on these results, they searched in the annotation of the genome of three Apicomplexa parasites for the existence of genes coding for proteins carrying KH motif. They identified about twenty of them which they classified into three groups according to their homologies, group 1 seeming the most promising in terms of transcription regulation potential. To reinforce this theoretical information, they did express the three proteins newly identified in the genome of P. falciparum, which they coupled to GFP in order to know their location in the infected red blood cell. One of these proteins co-localizes with the nucleus throughout the erythrocyte cycle. The authors conclude that it could be a new transcription factor.

The work presented by the authors is of good quality, well described and provides substantial data. The material and method part is well documented and allows a good understanding of how the data was obtained. The results are clearly presented and the discussion interesting even if the chapter lines 560-581 could be shortened.

I have two minor remarks

The first concerns the use of qPCR to study the association between PREBP and the promoter already identified. It would have been more appropriate to do RNAseq on the CHIPs. In addition to confirming the PREBP target, it could have revealed other targets on other parts of the genome and made it possible to identify a gene transcription cascade regulated by PREBP.

The second concerns the bioinformatics part of the manuscript. For example when we test on Plasmodb the "KH_1" search for P. falciparum (isolate 3D7), we find 5 genes as authors did but not exactely the same : the PF3d7_0302800 gene does not come out (because the denomination in interpro domain is "K Homology domain, type 1" and not "KH_1" as for the others) on the other hand we find the Pf3D7_1469300 gene which the authors do not talk about… is there an explanation for this?

Reviewer #2: Komaki-Yasuda and Kano profile Prx regulatory element-binding protein (PREBP) in P. falciparum blood stage parasites using various tools and approaches including parasite transgenic lines, confocal microscopy, and luciferase-based activity assays. The authors also perform a bioinformatic search for KH-domain proteins across various Apicomplexan parasites to identify new putative transcription factors. They then infer the roles of these domain-containing proteins as putative transcriptional regulators in Plasmodium spp. and beyond. They then use confocal imaging to profile one of these factors (PF3D7_0302800), showing its localization the nucleus P. falciparum blood stage parasites.

The study is related to- and builds on the authors’ previous work [17,18]. The methods chosen to profile PREBP are well thought out, however for PF3D7_0302800, a more thorough analysis could have been done. The researchers acknowledge that follow-up studies for this factor are necessary. The study has the potential to serve as a value resource to the malaria researcher community, but in the manuscript’s current format, the authors fall short of disseminating the findings in a manner that meet PLOS ONE's criteria for publication.

My major queries and concerns include the following:

Using confocal imaging, the authors follow the cellular localization of PREBP throughout the parasite's IDC. The results here support their previous findings showing the differential localization of PREBP using a complementary technique ([18] Figure 5). In Figure 1, the authors present representative images of the PREBP-GFP localization at different stages of the parasite's IDC. The current format of the data does not satisfy criterion #3 of PLOS ONE's publication requirements. To meet the replication standards, the authors should provide information as to how many parasites were assessed at each stage to allow them arrive at their conclusions. On Lines 270-278 the authors use terms such "throughout the cytoplasm" and "a particular strong region of green fluorescence...which coincided with the location of nuclear localization". These descriptions are rather subjective.The authors should perform co-localization analysis (for instance, using co-localization coefficients such as Pearson's correlation or Mander's split coefficients) to provide readers with quantitative support for your claims. Reporting both the number of parasites and the correlation assessment will add rigor to this analysis. In terms of controls, the authors generate a PfPREBP peptide antibody in [18] and use this antibody for ChIP analysis in the current study. I am curious as to why they do not use this antibody as a control in their confocal imaging analysis? This would provide visual support that addition of the GFP-tag to PREBP does not alter localization patterns. Presumably the co-localization coefficients would be the same when comparing the endogenous PREBP stained with anti-PREBP and a fluorescent secondary antibody to PREBP-GFP.

The authors use ChIP to assess the recruitment of PREBP on the cis-enhancer of the pf1-cys-prx gene. While this approach is powerful in assessing protein-DNA interactions, optimization is critical. The methods described pertinent to this analysis are not clear, making it very difficult for researchers to reproduce the protocol. For example, the authors report volumes in their methods section; however, these values are largely meaningless to the research community. It would be more informative to know what the quantity of material used for each of the steps in the workflow (i.e. ug, ng, mg, grams)? The authors should provide kit product numbers when applicable (i.e What is the Product number for the "The CHiP Assay Kit" (Thermo Fisher))? The authors fail to report any quality control steps for the assay. What were the positive and negative antibody controls used? A western blot showing that PREBP has been immunoprecipitated, along with the appropriate controls should be provided in the manuscript. Were positive and negative control loci used in RT-PCR? If so, the authors should provide the primer sequences. In Figure 2, the results related to the enrichment of target DNA using primers A and E are misleading. Presenting the data with error bars representing the standard deviation of 2 biological replicates is insufficient.The authors should overlay the raw data points. Likewise, the authors should perform a statistical test to compare relative enrichment of target DNA to primer set C. These modifications will satisfy #3 of PLOS ONE's publication criterion.

For the luciferase assay, the data represented in Figure 3 show: (1) the deletion of all 4 KH-domains negatively impacts pf1-cys-prx expression when the protein is truncated from the N-terminus. When all 4 KH-domains are deleted from a C-terminus, a decrease in pf1-cys-prx is not observed—unless the -cis enhancer in pf1-cys-prx is deleted. On lines 348-351, the authors indicated that "As a result, PREBP activity was not significantly reduced until it lacked 3 of 4 KH domains. Then, the activity was significantly decreased when it lacked all the putative KH domains (p < 0.01) (Fig 3C, right panel)"; however, this is not consistent with Figure 3. In the right panel they show that the activity is not significantly decreased unless the pf1-cys-prx -cis enhancer is deleted (i.e "I"). Please clarify these findings. Additionally, the authors should plot the individual data points on the graph (not just the bar plot +/- SD).

In the bioinformatic search for KH-domain containing proteins that may encode for transcription factors, the authors indicate that they removed several proteins that are orthologs of well-known proteins (Lines 401-402). Both PF3D7_0302800 (RNA-binding protein, putative) and PF3D7_1415300(RNA-binding protein Nova-1 putative) are annotated as RNA-binding proteins in PlasmoDB (release 61). The authors should clarify why these genes were not filtered out and are included in the search list. Additionally, the authors should provide the supplementary files with the genes that were included and excluded in the bioinformatic analysis. These lists will allow the research community to reproduce the workflow should it be required to re-run their search pipeline.

For the confocal microscopy assay assessing the localization of PF3D7_0302800, the authors should include the number of parasites assessed, and perform a co-localization analysis. Unlike PREBP [18], there is no supporting evidence for protein localization other than the confocal imaging data presented in the study. Since the PF3D7_0302800 is GFP-tagged, a western blotting of cytoplasmic and nuclear fractions of the parasite lysate would serve as complimentary method to support the authors' claim that the protein is localized to the nucleus over the course of its IDC.

Minor comments and questions:

Please adhere to PLOS' minimum data set definition for all analyses performed in this study. Along these lines, please indicate whether the plasmids generated in the study will be available, and if so, which repository the research community can find them. Same goes for the PREBP antibody and -GFP parasite lines, are these available to the research community? If so, how can they be obtained?

The authors indicate that PREBP can bind RNA (Line 19). When performing the ChIP analysis was an RNase used to digest RNA? Is it possible that the transcription factor could be binding nascent RNA or pre-mRNA in the nucleus? A bioanalyzer electropherogram for both DNA and for RNA would provide support for your ChIP approach.

Line 57: Please add the gene number (i.e PF3D7_XXXXXXX) when first mentioning pf1-cys-prx. Searching for pf1-cys-prx on PlasmoDB brings up four different genes (https://plasmodb.org/plasmo/app/search?q=pf1-cys-prx%20&documentType=gene&organisms=Plasmodium%20falciparum%203D7). Which one is correct?

Line 88: "thorough" should be "throughout".

Line 93: Instead of "some Apicomplexan parasites" it is better to state: "various Apicomplexan parasites".

Lines 104-171: The methods section describing the plasmid construction can be reduced significantly. Simply generate a table for the primers used and, in the text, direct the reader to this supplementary table in the event they would like to find these sequences.

Lines 383-385: The authors indicate that the left panel shows activity of the PREBP deletion series from the N terminus and the right panel shows that of the deletion series from the C terminus. However, "E:p1-10R-PREBP deltaC3" is a C-terminus deletion and it is being represented in the left panel instead of the right panel.

Please correct Figure Legend 3 to be consistent with what is described on Lines 383-385.

In the methods describing the bioinformatic analysis, please provide the database release # and date for each organism assessed. For example, Line 248 should read (PlasmoDB, Release #, day month year).

Lines 248-249: Were all Plasmodium spp. Toxoplasma spp. and Cryptosporidium spp. used in the screen? If not, please add strain information for those that were used

Line 258: which strains of the parasite species were used?

Line 390: I recommend using the word "putative" instead of "hypothetical" when describing the potential role of KH-domain containing proteins as transcription factors.

Line 439: What do the authors mean by "almost the same". It would be better to indicate the % overlap to provide the readers with a better indication of the similarity/dissimilarity between the approaches. A Venn Diagram would be useful here.

Line 451: Please provide clarification as to why ortholog group 1 was prioritized.

Lines 458 and 463: Please modify. As per their minimum data set definition, PLOS does not permit references to “data not shown.” PF3D7_1415300-GFP confocal data should be included as a supplementary file.

Line 508: "melts" is not the correct word here.

Lines 517: Please change pfGNC5 to italicized P. falciparum (Pf) GNC5; then, on lines 521 and 529 use PfGNC5.

Lines 532-525: Is this data generated in the current study? If so, please provide the alignment data and scores in a supplementary file.

Line 535-536: The statement here is incorrect. In PlasmoDB, PF3D7_0302800 is annotated as "RNA-binding protein, putative".

Line 538 should read "in the parasite nucleus throughout the ring, trophozoite, and schizont stages."

Line 562: State "were reported previously [40]" instead of "were reported [40] in 2015".

Line 576: What is a "typical" transcription factor?

Line 582-583: Please rephrase. There is extensive evidence showing AP2s as transcriptional regulators in Plasmodium species.

6. PLOS authors have the option to publish the peer review history of their article (what does this mean?). If published, this will include your full peer review and any attached files.

Reviewer #1: No

Reviewer #2: No

---

## [Author Response · Author response to Decision Letter 0]

25 Oct 2023

Response to Editor

Comment 1. Please ensure that your manuscript meets PLOS ONE's style requirements, including those for file naming. The PLOS ONE style templates can be found at

Response 1

We reviewed the PLOS ONE style requirements and adjusted the manuscript to align with them. Please let us know if there are any additional required modifications, and we will promptly address them.

Comment 2

Thank you for stating the following in the Acknowledgments Section of your manuscript:

“This work was 596 supported by JSPS KAKENHI Grant Numbers 17K08817 and 20K07473.”

“This work was supported by JSPS (Japan Society for the Promotion of Science) KAKENHI Grant Numbers 17K08817 and 20K07473 (to KKY).

The funder’s URL: https://www.jsps.go.jp/

Response 2

Thank you for bringing this to our attention. We have removed the funding-related text from the Acknowledgments section. Additionally, we have updated our Funding Statement as follows:

"Funding: This work was supported by JSPS (Japan Society for the Promotion of Science) KAKENHI Grant Numbers 17K08817 and 20K07473 (to KKY). The funder’s URL: https://www.jsps.go.jp/. This work was also supported by AMED (Japan Agency for Medical Research and Development) under Grant Number 23jk0210006h0001 (to SK). The funder’s URL: https://www.amed.go.jp. The funders had no role in study design, data collection and analysis, decision to publish, or preparation of the manuscript."

Comment 3

In your Data Availability statement, you have not specified where the minimal data set underlying the results described in your manuscript can be found. PLOS defines a study's minimal data set as the underlying data used to reach the conclusions drawn in the manuscript and any additional data required to replicate the reported study findings in their entirety. All PLOS journals require that the minimal data set be made fully available. For more information about our data policy, please see http://journals.plos.org/plosone/s/data-availability.

Response 3

In the revised manuscript, we have ensured that all relevant data necessary to understand and replicate the study findings are provided within the manuscript itself and the accompanying Supporting Information Files. No additional data repositories or URLs are required.

We kindly request that you update our Data Availability Statement to reflect this, indicating that all necessary data are included in the manuscript and its supplements.

Comment 4

We note that you have stated that you will provide repository information for your data at acceptance. Should your manuscript be accepted for publication, we will hold it until you provide the relevant accession numbers or DOIs necessary to access your data. If you wish to make changes to your Data Availability statement, please describe these changes in your cover letter and we will update your Data Availability statement to reflect the information you provide.

Response 4

Thank you for bringing this to our attention. We did not include data with repository information in our manuscript. All relevant data are within the manuscript and the Supporting Information files. The biological materials are available upon request. Therefore, we would like to request a modification to the Data Availability Statement.

If there are any additional steps required, please kindly advise us.

Thank you once again for your guidance.

Comment 5

PLOS requires an ORCID iD for the corresponding author in Editorial Manager on papers submitted after December 6th, 2016. Please ensure that you have an ORCID iD and that it is validated in Editorial Manager. To do this, go to ‘Update my Information’ (in the upper left-hand corner of the main menu), and click on the Fetch/Validate link next to the ORCID field. This will take you to the ORCID site and allow you to create a new iD or authenticate a pre-existing iD in Editorial Manager. Please see the following video for instructions on linking an ORCID iD to your Editorial Manager account: https://www.youtube.com/watch?v=_xcclfuvtxQ

Response 5

In response to your suggestion, I have created an ORCID iD and successfully linked it to my Editorial Manager account. Please verify the status of my ORCID iD, and inform me if there are any further requirements.

Comment 6

We note that you have included the phrase “data not shown” in your manuscript. Unfortunately, this does not meet our data sharing requirements. PLOS does not permit references to inaccessible data. We require that authors provide all relevant data within the paper, Supporting Information files, or in an acceptable, public repository. Please add a citation to support this phrase or upload the data that corresponds with these findings to a stable repository (such as Figshare or Dryad) and provide and URLs, DOIs, or accession numbers that may be used to access these data. Or, if the data are not a core part of the research being presented in your study, we ask that you remove the phrase that refers to these data.

Response 6

The data referred to as "data not shown" in the original manuscript are now presented in the Supporting Information. Figures S4, S5 and S6 in the revised manuscript correspond to these data.

Response to Reviewer 1

Comment 1

The first concerns the use of qPCR to study the association between PREBP and the promoter already identified. It would have been more appropriate to do RNAseq on the CHIPs. In addition to confirming the PREBP target, it could have revealed other targets on other parts of the genome and made it possible to identify a gene transcription cascade regulated by PREBP.

Response 1

The ChIP assay in this study was not performed under RNase-free conditions, making the analysis of RNA bound to PREBP impossible. However, to better understand the functions of PREBP, it is crucial to determine which RNA species bind to PREBP in the parasite cell. Therefore, in the future, experiments such as RNA-IP will be conducted to address this aspect.

Comment 2

The second concerns the bioinformatics part of the manuscript. For example when we test on Plasmodb the "KH_1" search for P. falciparum (isolate 3D7), we find 5 genes as authors did but not exactely the same: the PF3d7_0302800 gene does not come out (because the denomination in interpro domain is "K Homology domain, type 1" and not "KH_1" as for the others) on the other hand we find the Pf3D7_1469300 gene which the authors do not talk about… is there an explanation for this?

Response 2

The discrepancy between your search queries and results in the manuscript may be attributed to the fact that my ortholog analysis was based on search results obtained from older versions of the databases. 

To provide transparency regarding the search queries and results, we included a supplemental table that provides information on genes included and excluded from the bioinformatics analysis (Table S2).

Response to Reviewer 2

Major comments

Comment 1-1

Using confocal imaging, the authors follow the cellular localization of PREBP throughout the parasite's IDC. The results here support their previous findings showing the differential localization of PREBP using a complementary technique ([18] Figure 5). In Figure 1, the authors present representative images of the PREBP-GFP localization at different stages of the parasite's IDC. The current format of the data does not satisfy criterion #3 of PLOS ONE's publication requirements. To meet the replication standards, the authors should provide information as to how many parasites were assessed at each stage to allow them arrive at their conclusions. 

Response 1-1

At the time of the original manuscript, we utilized the Olympus FV1000 confocal laser microscope. However, we have since upgraded to the FV3000 model. Consequently, during the manuscript revision process, we acquired new images of all parasite cells at a higher resolution. Additionally, we conducted a fresh analysis of nuclear and GFP colocalization using an image analysis software program, quantifying colocalization through Pearson's correlation.

To reflect these changes, we have incorporated new images showing the subcellular distribution of PREBP as Figure 2, and images depicting the subcellular distribution of PF3D7_0302800 are now included as Figure 6 in the revised manuscript. Furthermore, we have introduced graphs illustrating the comparison of Pearson's correlations as a measure of colocalization across different developmental stages. These graphs can be found in Figures 2 and 7, accompanied by information on the number of cells analyzed. 

Furthermore, after performing a reanalysis using a high-resolution microscope, there were some slight modifications to the interpretation of PREBP cellular localization. In late schizonts, PREBP, which initially appeared to be located outside the nucleus, was observed as spots in the internal peripheral region of the nucleus. On the basis of the observations mentioned above, several modifications have been incorporated into the main text, as follows:

(Lines 20-22)

"PREBP was localized in the nucleus in the trophozoite and schizont stages, in which the expression of the target pf1-cys-prx was enhanced."

(Lines268-270)

"During the late schizont stage, green fluorescence corresponding to the localization of PREBP-GFP was observed as spots in the internal peripheral region of the nucleus."

Comment 1-2

On Lines 270-278 the authors use terms such "throughout the cytoplasm" and "a particular strong region of green fluorescence...which coincided with the location of nuclear localization". These descriptions are rather subjective. The authors should perform co-localization analysis (for instance, using co-localization coefficients such as Pearson's correlation or Mander's split coefficients) to provide readers with quantitative support for your claims. Reporting both the number of parasites and the correlation assessment will add rigor to this analysis.

Response 1-2

We performed a quantitative analysis of colocalization using ImageJ software program. The calculated Pearson’s correlation values are presented in the form of graphs in Figures 2 and 7 of the revised manuscript.

The interpretation of subcellular localization changed based on observations using a high-resolution microscope. Initially, we stated that PREBP moves outside the nucleus during the late schizont stage. However, a further analysis with a high-resolution microscope revealed that PREBP is located inside the nucleus, but not uniformly throughout the nucleus. Instead, its localization is restricted to a specific region within the nucleus.

(Refer to the response to comment 1-1 for details.)

Regarding PF3D7_0302800, during the ring and schizont stages, the nuclear localization and PF3D7_0302800 exhibited partial overlap, whereas colocalization was observed during the trophozoite stage. We have integrated these observations into the revised manuscript, as follows:

(Lines 29-31)

"One of the P. falciparum-derived factors, which were included in the putative ortholog group, was found to be localized at the nucleus in the trophozoite stage, indicating that it might be a novel transcription factor. "

(Lines 473-474)

"The localization of PF3D7_0302800-GFP localization matched the nuclear location in throughout the ring, the trophozoite stage and schizont stages (Fig 6). "

(Lines 480-482)

"These results suggested that PF3D7_0302800 could function as a transcription factor in the nucleus of trophozoites. "

(Lines 562-564)

" In the present study, the cellular localization of PF3D7_0302800 was analyzed, and it was found to be localized in the parasite nucleus during the trophozoite stage."

Comment 1-3

In terms of controls, the authors generate a PfPREBP peptide antibody in [18] and use this antibody for ChIP analysis in the current study. I am curious as to why they do not use this antibody as a control in their confocal imaging analysis? This would provide visual support that addition of the GFP-tag to PREBP does not alter localization patterns. Presumably the co-localization coefficients would be the same when comparing the endogenous PREBP stained with anti-PREBP and a fluorescent secondary antibody to PREBP-GFP.

Response 1-3

We appreciate your suggestion regarding the use of IFA with anti-PREBP antibodies in conjunction with confocal imaging analysis. Indeed, the use of both methods can offer complementary insights into protein localization. We opted not to perform IFA using anti-PREBP antibodies alongside PREBP-GFP fusion protein expression due to the advantages offered by the latter method. The use of PREBP-GFP fusion proteins allowed us to directly observe living cells, facilitating the examination of a larger number of cells more efficiently. Furthermore, our previous findings, as presented in the Western blot analysis of parasite cell fractions [18] using anti-PREBP antibodies, corroborate the localization patterns in this study. This consistency suggests that the fusion with GFP does not alter the localization pattern of PREBP and reinforces the reliability of our conclusions. We appreciate your understanding of this rationale.

Comment 2-1

The authors use ChIP to assess the recruitment of PREBP on the cis-enhancer of the pf1-cys-prx gene. While this approach is powerful in assessing protein-DNA interactions, optimization is critical. The methods described pertinent to this analysis are not clear, making it very difficult for researchers to reproduce the protocol. For example, the authors report volumes in their methods section; however, these values are largely meaningless to the research community. It would be more informative to know what the quantity of material used for each of the steps in the workflow (i.e. ug, ng, mg, grams)?

Response 2-1

In the original manuscript, the antibody quantities were already provided in µg. In the revised manuscript, we have made adjustments to include both the volume of cell lysate and the amount converted based on the number of cells. Furthermore, we have corrected the measurement unit for the amount of Dynabeads to mg.

Comment 2-2

The authors should provide kit product numbers when applicable (i.e What is the Product number for the "The ChIP Assay Kit" (Thermo Fisher))?

Response 2-2

In accordance with the comments, we have verified the current catalog number and manufacturer of the relevant kit, which is (Cat No. 17295, Merck-Millipore, Burlington, USA), and have included this information in the revised manuscript.

(Line 183)

Comment 2-3

The authors fail to report any quality control steps for the assay. What were the positive and negative antibody controls used?

Response 2-3

The ChIP assay included the use of normal mouse IgG as a negative control, and the corresponding data have been incorporated into Figure 3 of the revised manuscript. As for a positive control antibody, we did not utilize one because of the lack of available knowledge on an appropriate antibody positive control at the time of this study.

Comment 2-4

A western blot showing that PREBP has been immunoprecipitated, along with the appropriate controls should be provided in the manuscript.

Response 2-4

The chromatin immunoprecipitation products were subjected to a Western blot analysis to confirm the precipitation of PREBP. The corresponding data are presented in the supporting information as Figure S2.

Comment 2-5

Were positive and negative control loci used in RT-PCR? If so, the authors should provide the primer sequences.

Response 2-5

We did not employ positive or negative control loci in this RT-PCR, since we currently lack information on suitable control loci.

Comment 2-6

In Figure 2, the results related to the enrichment of target DNA using primers A and E are misleading. Presenting the data with error bars representing the standard deviation of 2 biological replicates is insufficient. The authors should overlay the raw data points. Likewise, the authors should perform a statistical test to compare relative enrichment of target DNA to primer set C. These modifications will satisfy #3 of PLOS ONE's publication criterion.

Response 2-6

Following your comment, we conducted a third experiment, and the current figure depicts the average and error bars from the results of the three experiments. Additionally, we have overlaid the raw data points.

The data sets obtained from Primer sets A, B, D, and E were compared with the dataset from Primer Set C, and as a result, statistical significance (p<0.01) was observed in all cases (indicated by asterisks on Figure 3 in the revised manuscript).

Comment 3

For the luciferase assay, the data represented in Figure 3 show: (1) the deletion of all 4 KH-domains negatively impacts pf1-cys-prx expression when the protein is truncated from the N-terminus. When all 4 KH-domains are deleted from a C-terminus, a decrease in pf1-cys-prx is not observed—unless the -cis enhancer in pf1-cys-prx is deleted. On lines 348-351, the authors indicated that "As a result, PREBP activity was not significantly reduced until it lacked 3 of 4 KH domains. 

Then, the activity was significantly decreased when it lacked all the putative KH domains (p < 0.01) (Fig 3C, right panel)"; however, this is not consistent with Figure 3. In the right panel they show that the activity is not significantly decreased unless the pf1-cys-prx -cis enhancer is deleted (i.e "I"). Please clarify these findings. Additionally, the authors should plot the individual data points on the graph (not just the bar plot +/- SD).

Response 3

First, in the section related to the comments, in the right-side graph of Figure 3C (Figure 4C in the revised manuscript), a correction was made to the annotation at the bottom of the graph. In the graph, the notation "I" (p1-10R PREBP ΔC3) was mistakenly labeled "E," and this has been rectified.

As indicated by "I", when all four KH domains are deleted (in the absence of cis-enhancer deletion), the luciferase expression of the pf1-cys-prx promoter was significantly decreased. This result aligns with the description in the main text. Data for the cis-enhancer deletion are shown in B', H', and I', demonstrating that the impact of cis-enhancer deletion is consistent with the effect of KH domain deletion.

In accordance with the comments, all raw data points have been added to the graph.

Comment 4

In the bioinformatic search for KH-domain containing proteins that may encode for transcription factors, the authors indicate that they removed several proteins that are orthologs of well-known proteins (Lines 401-402). Both PF3D7_0302800 (RNA-binding protein, putative) and PF3D7_1415300(RNA-binding protein Nova-1 putative) are annotated as RNA-binding proteins in PlasmoDB (release 61). The authors should clarify why these genes were not filtered out and are included in the search list.

 Additionally, the authors should provide the supplementary files with the genes that were included and excluded in the bioinformatic analysis. These lists will allow the research community to reproduce the workflow should it be required to re-run their search pipeline.

Response 4

A close examination of the actual analysis performed revealed the following exact situation. We excluded several proteins that are orthologs of well-known ribosomal RNA-associated proteins (rather than RNA-binding proteins), and those not predicted to contain KH domains. The text has been revised to emphasize this.

(Lines 415-417)

"Then, we removed several proteins that are orthologs of well-known ribosomal RNA-related proteins from the protein lists indicated as results of the search (S2 Table). "

Furthermore, in response to feedback from reviewers, we included a supplemental table that provides information on genes included and excluded from the bioinformatics analysis.

(Table S2)

Comment 5

For the confocal microscopy assay assessing the localization of PF3D7_0302800, the authors should include the number of parasites assessed, and perform a co-localization analysis. Unlike PREBP [18], there is no supporting evidence for protein localization other than the confocal imaging data presented in the study. Since the PF3D7_0302800 is GFP-tagged, a western blotting of cytoplasmic and nuclear fractions of the parasite lysate would serve as complimentary method to support the authors' claim that the protein is localized to the nucleus over the course of its IDC.

Response 5

We have repeated confocal laser microscopy imaging to observe the cellular localization of PF3D7_0302800-GFP, similar to our approach for PREBP. The newly acquired images are now presented in Figure 6 of the revised manuscript. As with PREBP, we include a graphical representation of the quantitative results of the colocalization analysis in Figure 7. The number of cells analyzed was indicated in the graph.

Although Western blotting serves as a valuable tool to validate microscopic localization observations, we also plan to perform more comprehensive analyses, including the ChIP assay for PF3D7_0302800, in our upcoming research efforts.

Minor comments

Comment 1

Please adhere to PLOS' minimum data set definition for all analyses performed in this study. Along these lines, please indicate whether the plasmids generated in the study will be available, and if so, which repository the research community can find them. Same goes for the PREBP antibody and -GFP parasite lines, are these available to the research community? If so, how can they be obtained?

Response 1

According to PLOS' minimal data set definition, in the revised manuscript, the relevant data are within the manuscript and the Supporting Information files. We did not deposit any plasmids, antibodies, or cell lines generated in this study into open resources. Thus, biological materials are available on request. Therefore, we have asked the editor to update this statement accordingly.

Comment 2

The authors indicate that PREBP can bind RNA (Line 19). When performing the ChIP analysis was an RNase used to digest RNA? Is it possible that the transcription factor could be binding nascent RNA or pre-mRNA in the nucleus? A bioanalyzer electropherogram for both DNA and for RNA would provide support for your ChIP approach.

Response 2

As we noted in our response to the first comment from Reviewer 1, the ChIP assay in this study was not performed under RNase-free conditions, making the analysis of RNA bound to PREBP impossible. However, to better understand the functions of PREBP, it is crucial to determine which RNA species bind to PREBP in the parasite cell. Therefore, experiments such as RNA-IP in the future will be conducted to address this aspect.

Comment 3

Line 57: Please add the gene number (i.e PF3D7_XXXXXXX) when first mentioning pf1-cys-prx. Searching for pf1-cys-prx on PlasmoDB brings up four different genes (https://plasmodb.org/plasmo/app/search?q=pf1-cys-prx%20&documentType=gene&organisms=Plasmodium%20falciparum%203D7). Which one is correct?

Response 3

The gene ID for the pf1-cys-prx gene is PF3D7_0802200. This number was added in the first mention of pf1-cys-prx in the revised manuscript. (Lines 58-59)

Comment 4

Line 88: "thorough" should be "throughout".

Response 4

The word "through" has been corrected in the revised manuscript to "throughout". 

(Line 89)

Comment 5

Line 93: Instead of "some Apicomplexan parasites" it is better to state: "various Apicomplexan parasites".

Response 5

The term "some Apicomplexan parasites" has been corrected to "various Apicomplexan parasites".

(Line 94)

Comment 6

Lines 104-171: The methods section describing the plasmid construction can be reduced significantly. Simply generate a table for the primers used and, in the text, direct the reader to this supplementary table in the event they would like to find these sequences.

Response 6

In the Methods section, we removed detailed primer sequence data and instead added a table (Table S1) compiling a list of primers as supplemental data.

Comment 7

Lines 383-385: The authors indicate that the left panel shows activity of the PREBP deletion series from the N terminus and the right panel shows that of the deletion series from the C terminus. However, "E: p1-10R-PREBP deltaC3" is a C-terminus deletion and it is being represented in the left panel instead of the right panel.

Response 7

In the right-side graph of Figure 3C (Figure 4C in the revised manuscript), a correction was made to the annotation at the bottom of the graph. In the graph, the notation "I" (p1-10R PREBP ΔC3) was mistakenly labeled "E," and this has been rectified.

Comment 8

Please correct Figure Legend 3 to be consistent with what is described on Lines 383-385.

Response 8

Please refer to the response to comment 3.

Comment 9

In the methods describing the bioinformatic analysis, please provide the database release # and date for each organism assessed. For example, Line 248 should read (PlasmoDB, Release #, day month year).

Response 9

In the revised manuscript, the database release # and date for each organism were provided.

(Lines 236-239)

" To search for candidate proteins that contain KH-domains in Apicomplexan organisms, we first screened the genome database for each Plasmodium (PlasmoDB 28, release 30 March 2016) [32], Toxoplasma (ToxoDB 28, release 30 March 2016) [33], and Cryptosporidium (CryptoDB 28, release 30 March 2016) [34] by the word “KH”."

Comment 10

Lines 248-249: Were all Plasmodium spp. Toxoplasma spp. and Cryptosporidium spp. used in the screen? If not, please add strain information for those that were used.

Response 10

To make this point clear, I added the following sentence in the relevant section:

(Lines 240-241)

"Subsequently, proteins from the P. falciparum 3D7 strain, T. gondii TG1 strain, and C. parvum Iowa II strain, identified through this search, were selected for further analysis."

Comment 11

Line 258: which strains of the parasite species were used?

Response 11

We added the following sentence:

(Lines 250-252)

"Proteins from the P. falciparum 3D7 strain, T. gondii TG1 strain, and C. parvum Iowa II strain, identified through this search, were selected."

Comment 12

Line 390: I recommend using the word "putative" instead of "hypothetical" when describing the potential role of KH-domain containing proteins as transcription factors.

Response 12

I have corrected all instances of the expression "putative KH domain" in the manuscript to "hypothetical KH domain."

(Lines 95, 360, 366, 368, 380, 407, 415, 423, 507, 562, 570, 575 and 797)

Comment 13

Line 439: What do the authors mean by "almost the same". It would be better to indicate the % overlap to provide the readers with a better indication of the similarity/dissimilarity between the approaches. A Venn Diagram would be useful here.

Response 13

In accordance with the comment, the following revision to the relevant section:

(Lines 453-459)

" Among the putative homologs identified by screening with candidate proteins of Group 1 (Fig 5), 70% were confirmed as actual members of Group 1 (S3 Fig). Similarly, during the search for putative homologs by screening proteins from Group 2 (Fig 5), it was discovered that 85.7% indeed belonged to Group 2, demonstrating a strong similarity (S3 Fig). Thus, in relation to putative orthologs within Group 1 and Group 2, the mutual BLASTP search provided further support for the results obtained from BLASTClust." 

Furthermore, I have added pie charts to the S3Fig to visually represent the results of the mutual BLASTP search.

Comment 14

Line 451: Please provide clarification as to why ortholog group 1 was prioritized.

Response 14

In our current analysis, we used PREBP as a well-established transcription factor to initiate an exploration of analogous factors within the malaria parasite. In this study, we prioritized the analysis of factors within ortholog group 1, expecting that they may share certain characteristics with PREBP orthologs.

Comment 15

Lines 458 and 463: Please modify. As per their minimum data set definition, PLOS does not permit references to “data not shown.” PF3D7_1415300-GFP confocal data should be included as a supplementary file.

Response 15

Similarly to the cases of PREBP and PF3D7_0302800, we also acquired confocal microscopy images for cellular localization of PF3D7_1415300-GFP. We have included the image data along with the results of quantitative analysis of colocalization between PF3D7_1415300-GFP and the nucleus in the supporting information.

(Figure S4 and Figure S5)

On the basis of the results of the quantitative colocalization analysis, we have made relevant revisions to the sentence, as follows:

(Lines 478-480)

"An analysis of colocalization between PF3D7_1415300-GFP and the nucleus was conducted; however, no consistent pattern of colocalization was observed (S4 Fig, S5 Fig)."

Comment 16

Line 508: "melts" is not the correct word here.

Response 16

The "melts" have been corrected to "dissociates" in the revised manuscript.

(Line 532)

Comment 17

Lines 517: Please change pfGNC5 to italicized P. falciparum (Pf) GNC5; then, on lines 521 and 529 use PfGNC5.

Response 17

Following the comment, the initial occurrence of "PfGCN5, histone acetyl-transferase" was changed to "italicized P. falciparum (Pf) GNC5, a histone acetyl-transferase" (Lines 541-542) and subsequently referred to it as "PfGCN5."

(Lines 545 and 554)

Comment 18

Lines 532-525: Is this data generated in the current study? If so, please provide the alignment data and scores in a supplementary file.

Response 18

We performed a BLAST search in the PlasmoDB gene database to identify conserved homologous proteins within the genus Plasmodium for PF3D7_0302800, PF3D7_0605100, and PF3D7_1415300. This search yielded homologs conserved across various Plasmodium species. Detailed alignment scores derived from the BLAST results are provided in Table S3.

Comment 19

Line 535-536: The statement here is incorrect. In PlasmoDB, PF3D7_0302800 is annotated as "RNA-binding protein, putative".

Response 19

According to the comment, the relevant sentence was rewritten as follows:

" Currently, PF3D7_0302800 is annotated as "RNA-binding protein, putative" in PlasmoDB (release 65), and contains one hypothetical KH domain."

Comment 20

Line 538 should read "in the parasite nucleus throughout the ring, trophozoite, and schizont stages."

Response 20

Regarding this point, after conducting a more comprehensive analysis of PF3D7_0302800 cellular localization, we found that it was located within the nucleus during the trophozoite stage. 

(See the reply to comment No. 1-2 for details)

As a result, the sentence in question was revised as follows:

(Lines 562-564) 

"In the present study, the cellular localization of PF3D7_0302800 was analyzed, and it was found to be localized in the parasite nucleus during the trophozoite stage."

Comment 21

Line 562: State "were reported previously [40]" instead of "were reported [40] in 2015".

Response 21

Based on the reviewer's comment, "were reported [40] in 2015" was revised to "were previously reported [40]." 

(Line 587)

Comment 22

Line 576: What is a "typical" transcription factor?

Response 22

We believed that "typical" transcription factors often possess a DNA-binding domain that recognizes and binds to specific DNA sequences (often referred to as cis-regulatory elements or transcription factor binding sites) within the promoter or enhancer regions of target genes, and they also have domains that interact with the transcriptional machinery, either enhancing or inhibiting gene transcription. However, considering the presence of AP2, we have revised the original expression as follows:

(Lines 601-602)

"In Apicomplexa, only a few typical transcription factors have been found, multiple functions may be concentrated in a few nucleic acid-binding proteins, and future studies are needed to determine their evolutionary significance."

Comment 23

Line 582-583: Please rephrase. There is extensive evidence showing AP2s as transcriptional regulators in Plasmodium species.

Response 23

According to the comment, the relevant section has been rewritten as follows:

(Lines 607-609)

"To date, despite extensive evidence supporting AP2s as transcriptional regulators in Plasmodium species, the detailed mechanisms of transcriptional regulation in Plasmodium and other Apicomplexan organisms are still not fully understood."

---

## [Decision Letter · Decision Letter 1]

6 Dec 2023

PONE-D-23-00650R1The RNA-binding KH-domain in the unique transcription factor of the malaria parasite is responsible for its transcriptional regulatory activityPLOS ONE

Dear Dr. Komaki-Yasuda,

Thank you for submitting your manuscript to PLOS ONE. After careful consideration, we feel that it has merit but does not fully meet PLOS ONE’s publication criteria as it currently stands. Therefore, we invite you to submit a revised version of the manuscript that addresses the points raised during the review process. I have marked you revised manuscript down for minor revision to allow you to correct the small number of grammatical errors, as they is no proof reading stage with PONE. These are listed below:

N 16: “The PREBP had been shown” should be “PREBP has been shown..”

LN 18: “The PREBP…” should be “PREBP…”

LN 27-28: “Bioinformatic analysis suggested…” should be “Bioinformatic analysis revealed…”

LN 75: “…PREBP was expected…” should be “…PREBP was predicted from…”

LN 84: “The finding of PREBP suggested…” should be “ These findings suggest…”

LN 95: “unknown” should be “putative”

LN 371: “(see Fig 4A)” should be “(Fig 4A)”

LN 409-410: “…organisms, unknown KH-domain containing proteins might act as unknown transcription factors” should be “…organisms, KH-domain containing proteins might act as transcription factors.”

LN 611-612: “…in the Apicomplexan…” should be “…in Apicomplexan…”Please ensure that your decision is justified on PLOS ONE’s publication criteria and not, for example, on novelty or perceived impact.

We look forward to receiving your revised manuscript.

Kind regards,

Gordon Langsley

Academic Editor

PLOS ONE

Journal Requirements:

Reviewers' comments:

Reviewer's Responses to Questions

**Comments to the Author**

1. If the authors have adequately addressed your comments raised in a previous round of review and you feel that this manuscript is now acceptable for publication, you may indicate that here to bypass the “Comments to the Author” section, enter your conflict of interest statement in the “Confidential to Editor” section, and submit your "Accept" recommendation.

Reviewer #2: All comments have been addressed

2. Is the manuscript technically sound, and do the data support the conclusions?

Reviewer #2: Yes

3. Has the statistical analysis been performed appropriately and rigorously? 

Reviewer #2: Yes

4. Have the authors made all data underlying the findings in their manuscript fully available?

Reviewer #2: Yes

5. Is the manuscript presented in an intelligible fashion and written in standard English?

Reviewer #2: Yes

6. Review Comments to the Author

Reviewer #2: The resubmission of the manuscript entitled “The RNA-binding KH-domain in the unique transcription factor of the malaria parasite is responsible for its transcriptional regulatory activity" was submitted by Komaki-Yasuda and Kano in response to two reviewer comments and recommendations. This manuscript is focused on understanding the role of KH-domains in transcriptional regulation of Plasmodium falciparum and potentially other Apicomplexans.

The amended manuscript has many strengths. The authors both re-analyze, and generate additional data to better support their claims. The resubmission is highly responsive to the reviewer comments. Based upon the evaluation of scientific and technical merit, this manuscript is acceptable for publication in its current format.

In terms of the general readability and accessibility of the manuscript: prior to publication, I highly recommend that the authors (perhaps with the guidance of editorial team at PLOS one) modify the title of their manuscript to make it more fitting to the body of work presented. For example, the authors argue that the KH-domain is important for gene regulation at the DNA level (i.e. as a transcription factor). The usage of “RNA-binding” in the title is somewhat confusing and takes away from the authors’ argument. Likewise, the use of “the unique transcription factor” in the title is very vague. Consider using the specific name of the transcription factor(s) that you are referring (i.e. PREBP etc). A more specific title will facilitates the retrieval of the paper from bibliographic databases.

Some grammatical errors that should be fixed prior to publication:

LN 16: “The PREBP had been shown” should be “PREBP has been shown..”

LN 18: “The PREBP…” should be “PREBP…”

LN 27-28: “Bioinformatic analysis suggested…” should be “Bioinformatic analysis revealed…”

LN 75: “…PREBP was expected…” should be “…PREBP was predicted from…”

LN 84: “The finding of PREBP suggested…” should be “ These findings suggest…”

LN 95: “unknown” should be “putative”

LN 371: “(see Fig 4A)” should be “(Fig 4A)”

LN 409-410: “…organisms, unknown KH-domain containing proteins might act as unknown transcription factors” should be “…organisms, KH-domain containing proteins might act as transcription factors.”

LN 611-612: “…in the Apicomplexan…” should be “…in Apicomplexan…”

7. PLOS authors have the option to publish the peer review history of their article (what does this mean?). If published, this will include your full peer review and any attached files.

Reviewer #2: No

---

## [Author Response · Author response to Decision Letter 1]

6 Dec 2023

Dear Editor,

We would like to express our gratitude to you and the reviewer for the thorough review of our manuscript. We are pleased to learn that our manuscript is deemed suitable for publication in PLOS ONE following minor grammatical corrections.

Kindly confirm that all the points listed below have been addressed in the revised manuscript in accordance with your suggestions:

LN 16: “The PREBP had been shown” should be “PREBP has been shown..”

LN 18: “The PREBP…” should be “PREBP…”

LN 27-28: “Bioinformatic analysis suggested…” should be “Bioinformatic analysis revealed…”

LN 75: “…PREBP was expected…” should be “…PREBP was predicted from…”

LN 84: “The finding of PREBP suggested…” should be “ These findings suggest…”

LN 95: “unknown” should be “putative”

LN 371: “(see Fig 4A)” should be “(Fig 4A)”

LN 409-410: “…organisms, unknown KH-domain containing proteins might act as unknown transcription factors” should be “…organisms, KH-domain containing proteins might act as transcription factors.”

LN 611-612: “…in the Apicomplexan…” should be “…in Apicomplexan…”

Regarding the reviewer's suggestion to modify the title, we believe in the significance of the factor with RNA-binding being involved in transcriptional regulation through DNA. Therefore, we kindly request your understanding in retaining the current title.

We have thoroughly reviewed the reference list and can confirm that it is complete, correct, and that no paper has been retracted.

Thank you very much.

Sincerely yours,

Kanako Komaki-Yasuda

---

## [Editor Report · Decision Letter 2]

8 Dec 2023

The RNA-binding KH-domain in the unique transcription factor of the malaria parasite is responsible for its transcriptional regulatory activity

PONE-D-23-00650R2

Dear Dr. Kanako Komaki-Yasuda,

We’re pleased to inform you that your manuscript has been judged scientifically suitable for publication and will be formally accepted for publication once it meets all outstanding technical requirements.

Kind regards,

Gordon Langsley

Academic Editor

PLOS ONE
---

## [Editor Report · Acceptance letter]

13 Dec 2023

PONE-D-23-00650R2 

PLOS ONE

Dear Dr. Komaki-Yasuda, 

I'm pleased to inform you that your manuscript has been deemed suitable for publication in PLOS ONE. Congratulations! Your manuscript is now being handed over to our production team.

Kind regards, 

on behalf of

Dr. Gordon Langsley 

Academic Editor

PLOS ONE